# The phosphatidylinositol-transfer protein Nir3 promotes PI(4,5)P$_2$ replenishment in response to TCR signaling during T cell development and survival

Wen Lu[1], Ynes A. Helou[1,5], Krishna Shrinivas [2], Jen Liou [3], Byron B. Au-Yeung [4] & Arthur Weiss [1] ✉

Hydrolysis of phosphatidylinositol 4,5-bisphosphate (PIP$_2$) by phospholipase C-γ (PLCγ1) represents a critical step in T cell antigen receptor (TCR) signaling and subsequent thymocyte and T cell responses. PIP$_2$ replenishment following its depletion in the plasma membrane (PM) is dependent on delivery of its precursor phosphatidylinositol (PI) from the endoplasmic reticulum (ER) to the PM. We show that a PI transfer protein (PITP), Nir3 (*Pitpnm2*), promotes PIP$_2$ replenishment following TCR stimulation and is important for T cell development. In *Nir3*$^{-/-}$ T lineage cells, the PIP$_2$ replenishment following TCR stimulation is slower. *Nir3* deficiency attenuates calcium mobilization in double-positive (DP) thymocytes in response to weak TCR stimulation. This impaired TCR signaling leads to attenuated thymocyte development at TCRβ selection and positive selection as well as diminished mature T cell fitness in *Nir3*$^{-/-}$ mice. This study highlights the importance of PIP$_2$ replenishment mediated by PITPs at ER-PM junctions during TCR signaling.

T lymphocytes play a central role in the adaptive immune response, providing protection from pathogens and malignant transformation. Upon engagement by their cognate antigens, TCRs trigger downstream signaling events that elicit robust T cell activation. Signaling by the TCR and the pre-TCR are also critically important for passage and maturation of thymocytes through development signaling checkpoints, including TCRβ selection, positive selection and negative selection. Compared with mature T cells, DP thymocytes are inherently more sensitive to low-potency ligands to elicit productive signals in response to weak interactions with self-peptide-MHC (pMHC) complexes on thymic stromal cells during positive selection[1]. The TCR sensitivity difference between immature and mature cells suggests the existence of TCR signaling components expressed selectively in thymocytes that influence TCR signaling during T cell development.

A key downstream feature of TCR signaling is elevation of intracellular free calcium concentration ($[Ca^{2+}]_i$)[2]. TCR-induced $[Ca^{2+}]_i$ elevation is a consequence of the metabolism of PIP$_2$ into inositol trisphosphate (IP$_3$) and diacylglycerol (DAG) through the action of tyrosine-phosphorylated PLCγ1 (refs. 3,4). PIP$_2$, an essential substrate for TCR-induced calcium signaling, only represents a minor lipid species in the inner leaflet of the PM[5]. Within one minute of stimulation with anti-CD3 antibodies, the level of PIP$_2$ in human T cells can decrease

[1]Rosalind Russell and Ephraim P. Engleman Rheumatology Research Center, Departments of Medicine and of Microbiology and Immunology, University of California, San Francisco, San Francisco, CA, USA. [2]NSF-Simons Center for Mathematical & Statistical Analysis of Biology, Harvard University, Cambridge, MA, USA. [3]Department of Physiology, University of Texas Southwestern Medical Center, Dallas, TX, USA. [4]Division of Immunology, Lowance Center for Human Immunology, Department of Medicine, Emory University, Atlanta, GA, USA. [5]Present address: Clade Therapeutics, Cambridge, MA, USA. ✉e-mail: Arthur.Weiss@ucsf.edu

by more than 50%. At the same time, TCR stimulation accelerates the production of $PIP_2$ from its precursor PI to replenish the PM $PIP_2$ pool[6,7]. To date, much of the knowledge about $PIP_2$ homeostasis after TCR stimulation has been studied in immortalized human T cell lines. It is not clear how PM $PIP_2$ levels are regulated in developing primary thymocytes or mature peripheral T cells.

PI is primarily synthesized in the ER and needs to be transferred to the PM for $PIP_2$ synthesis. Membrane-associated PITPs are major regulators of PM $PIP_2$ levels by transferring PI from the ER to the PM at ER-PM junctions, where the ER forms close contacts with the PM[8]. In mammalian genomes, there are three members of the membrane-associated PITP family. Nir1 (encoded by *Pitpnm3*) lacks the PITP domain and functions as a scaffold protein[9]. Both Nir2 (encoded by *Pitpnm1*) and Nir3 (encoded by *Pitpnm2*) transfer PI from the ER to the PM in response to $PIP_2$ hydrolysis; they each have an intact N-terminal PITP domain and membrane-targeting C-terminal domains, which bind to phosphatidic acid (PA) and DAG[10,11]. Compared with Nir2, the C-terminal domains of Nir3 are more sensitive to DAG and PA production in the PM. Thus, Nir3 can translocate to ER-PM junctions at low DAG and PA levels in the PM generated in response to weak receptor stimulation. On the other hand, the PITP domain of Nir3 is less potent in transferring PI and replenishing PM $PIP_2$[10]. Nir2 and Nir3 have been demonstrated to regulate $PIP_2$ homeostasis and $[Ca^{2+}]_i$ elevation in response to angiotensin II receptor and histamine receptor stimulation[10,12]. However, their roles in TCR signaling are unknown.

Here, we study Nir3, a membrane-associated PITP conserved among all vertebrates and its involvement in the thymocyte selection. Nir3 is most highly expressed in late double-negative (DN) and DP thymocytes and is downregulated after positive selection. We show that Nir3 promotes both TCRβ selection and positive selection in thymocytes. Nir3 also promotes mature T cell survival in the periphery. Mechanistically, Nir3 is required for $PIP_2$ replenishment and TCR-induced calcium signaling in response to weak TCR stimulation.

## Results

### Expression and evolution of Nir3 in T cell development

Across human tissues, *Nir2* transcripts are ubiquitously expressed, whereas *Nir3* transcripts are highly enriched in the thymus (Fig. 1a)[13]. In mouse T cell subsets, *Nir3* transcripts are elevated in late DN to DP stages and are decreased in single-positive (SP) thymocytes and mature T cells. Interestingly, *Nir3* transcripts are substantially elevated as thymocytes progress through TCRβ selection (DN3b) and positive selection (DP_Post). On the other hand, *Nir2* transcripts are relatively low in late DN to DP stages and are elevated in more mature thymocytes and mature T cells (Fig. 1b). Moreover, we determined the relative protein abundance of Nir2 and Nir3 in mouse thymocytes and mouse peripheral T cells by immunoblot analysis with additional normalization controls in which Nir2-mCherry or Nir3-mCherry fusion proteins were ectopically expressed in HEK293 cells. Nir3 protein is more highly expressed in thymocytes, whereas Nir2 protein is more abundant in peripheral T cells (Fig. 1c,d). These data are consistent with published quantitative mass spectrometry analysis of T cell proteomes, which demonstrated Nir2 protein is more abundant than Nir3 protein in peripheral T cells[14].

Given their sequence similarity and conserved domain organization, Nir2 and Nir3 likely arose as a result of a gene duplication event from a common ancestor during evolution. To investigate the evolutionary history of these two proteins, we identified orthologs across multiple animal lineages and developed a phylogenetic reconstruction analysis. From the reconstructed phylogenetic tree, we found that the most recent common ancestor (MRCA) of Nir2 and Nir3 was likely located at the base of all vertebrates (Extended Data Fig. 1). It is interesting to note that the modern adaptive immune system arose at a similar time in jawed vertebrates, concordant with the duplication and specialization of multiple gene families with prominent roles in TCR signaling, including Zap70/Syk and Lck/Lyn[15]. Moreover, our analysis suggests that shortly after the duplication of Nir2 and Nir3, the Nir2-like

paralogs were lost in fish and many amphibians. As a result, all extant fish carry Nir3, but not Nir2, in their genomes, the origin of which is not well understood (Extended Data Fig. 1).

In humans, several single-nucleotide polymorphisms of *Nir3* have been significantly associated with several immune-related diseases, including asthma, eczema, multiple sclerosis and type II diabetes by genome-wide association studies (GWAS Catalog, Supplementary Table 1). One of the *Nir3* single-nucleotide polymorphisms, rs7132277 has been known as a non-MHC susceptibility locus associated with multiple sclerosis at a genome-wide significance level[16]. Collectively, our findings that Nir3 was separated from Nir2 at a similar time as emergence of the modern adaptive immune system and is associated with immune-related diseases in human highly suggest a role for Nir3 in T cell development and function.

### Thymocyte maturation is accompanied by an increase in $PIP_2$ levels

We have previously reported that thymocyte positive selection is accompanied by a gradual increase in both basal free intracellular calcium concentrations ($[Ca^{2+}]_i$) and TCR-induced $[Ca^{2+}]_i$, which reflects stronger TCR signaling[17]. Given the critical role of $PIP_2$ in TCR-induced calcium signaling, we hypothesized that thymocyte maturation is also accompanied by an increase in $PIP_2$ levels. To test this hypothesis, we used a $PIP_2$-specific monoclonal antibody to examine $PIP_2$ levels in thymocyte subsets. Relative to DP thymocytes, our analyses revealed significantly elevated $PIP_2$ levels in SP thymocytes, with the highest $PIP_2$ levels in CD8SP cells (Fig. 2a–c).

DN thymocytes can be separated into four sequential developmental stages (DN1–DN4) based on their expression of CD44 and CD25 (Fig. 2d). TCRβ selection takes place at the DN3 stage, which can be further divided into the preselection DN3a stage and the postselection DN3b stage (Fig. 2e)[18]. Among the different DN subsets, $PIP_2$ levels in the cells were elevated as DN thymocytes progressed though the TCRβ-selection checkpoint. The $PIP_2$ level peaked at the DN3b stage and was slightly diminished in cells that had progressed into the DN4 stage (Fig. 2f,g).

After the DN4 stage, thymocytes upregulate CD4 and CD8 co-receptors and enter the DP stage. Throughout the DP stage, thymocytes undergoing positive selection are present in graded states of maturation, which can be visualized by cell surface levels of CD5 and TCRβ[19]. Subdividing DP thymocytes based on their CD5 and TCRβ levels revealed gradual elevations in $PIP_2$ levels as thymocytes progressed though positive selection, mirroring the elevated TCRβ levels (Fig. 2h–k). The gradually elevated $PIP_2$ levels are also consistent with previously observed elevations in basal and TCR-induced $[Ca^{2+}]_i$ in maturing DP thymocytes[17]. These data supported our hypothesis that an increase in $PIP_2$ levels is accompanied by elevated TCR expression and correlates with the basal $[Ca^{2+}]_i$ increase during thymocyte maturation.

### Nir3 promotes TCRβ selection

The high expression of Nir3 in thymocytes suggested a role for Nir3 in thymocyte development. To study Nir3 and its role in thymocyte development, a Nir3 genetic knockout (*Nir3*[−/−]) mouse was generated using CRISPR gene editing. Overall, *Nir3*[−/−] mice were healthy and fertile and born at the expected Mendelian ratios. In the thymus, a slight but statistically significant decrease in the percentage of CD4SP thymocytes was found in *Nir3*[−/−] mice, hinting at a potential defect in thymocyte development (Fig. 3a,b). In DN thymocytes, *Nir3*[−/−] mice developed fewer DN4 cells than wild-type (WT) control mice, suggesting a partial impairment of thymocyte development in the transition from the DN3 to the DN4 stage (Fig. 3c,d). At the DN3 stage, based on expression of intracellular TCRβ (icTCRβ) and the cell activation marker CD27, there were 50% fewer postselection DN3b cells in *Nir3*[−/−] mice than in WT mice (Fig. 3e,f). The *Nir3*[−/−] DN3 cells express substantially less CD27, which suggests Nir3 is an important regulator for thymocyte passage through the TCRβ-selection checkpoint (Fig. 3g).

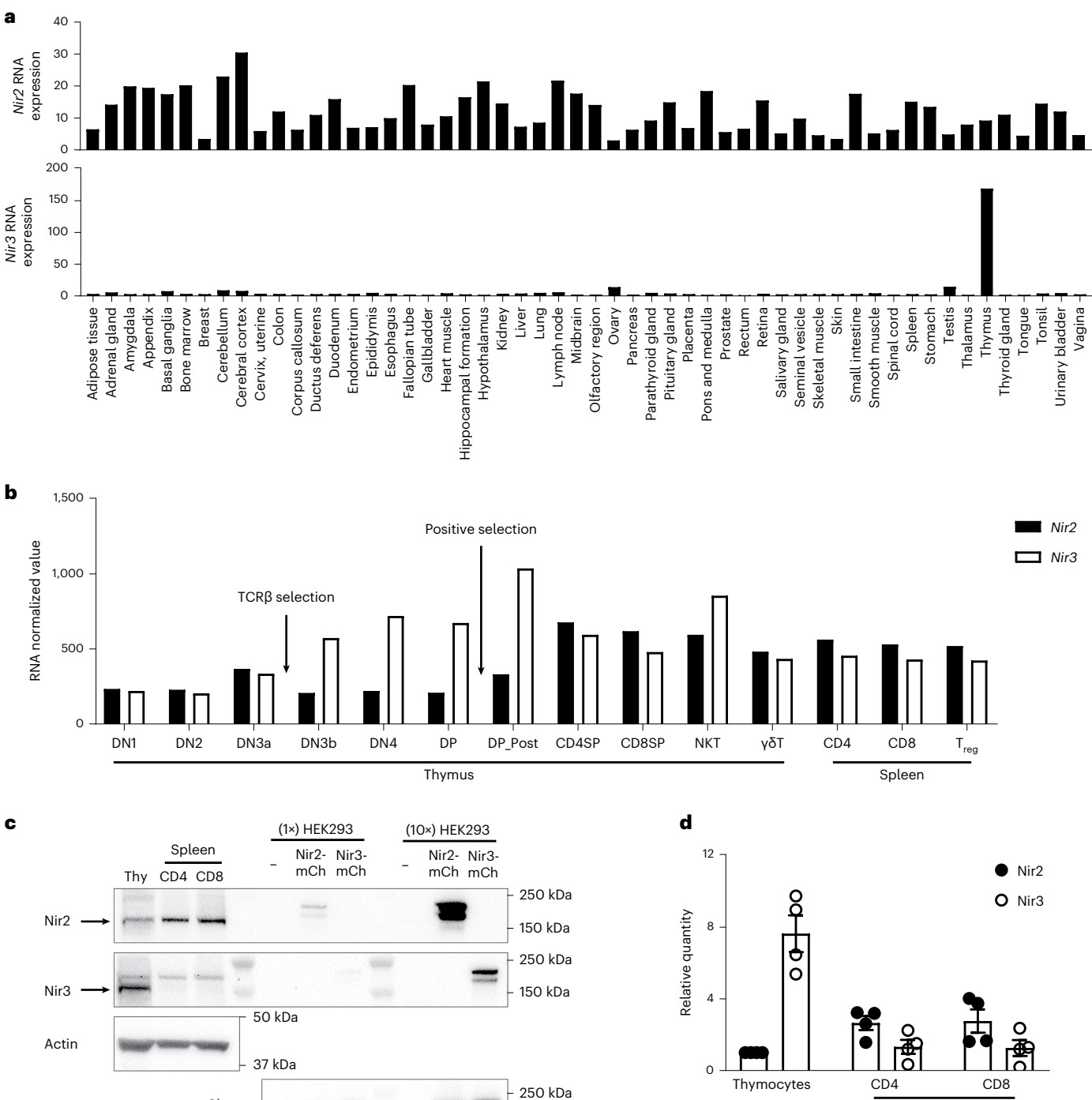

**Fig. 1 | Tissue expression of *Nir2* and *Nir3*. a**, Normalized mRNA expression (nTPM) levels of *PITPNM1* (*Nir2*) and *PITPNM2* (*Nir3*) for 55 human tissue types, derived from the Human Protein Atlas database. **b**, Normalized expression values (generated from Immunological Genome Project datasets) of *Pitpnm1* (*Nir2*) and *Pitpnm2* (*Nir3*) in selected T cell subsets in thymus and spleen. **c,d**, Protein expression of Nir2 and Nir3 in thymocytes (Thy) and splenic T cells. **c**, Representative expression of Nir2 and Nir3 was assessed by immunoblots. Ectopic expression of Nir2-mCherry (Nir2-mCh) and Nir3-mCherry (Nir3-mCh)

in HEK293 cells was included for normalization. **d**, The band intensities of Nir2 and Nir3 from four independent experiments similar to those shown in panel c were quantified and normalized to Nir2 expression in thymocytes. The relative abundance of Nir2 and Nir3 in each sample was first normalized to actin and was subsequently normalized to ectopic expression of Nir2-mCh or Nir3-mCh in HEK293 cells. Data are shown as the mean ± standard error of the mean (s.e.m.) and are collection of four independent experiments.

## Positive selection is impaired in *Nir3*$^{-/-}$ mice

To study the role of Nir3 in positive selection of DP thymocytes, we subdivided DP thymocytes into five populations based on their expression of CD5 and TCRβ. Mildly decreased percentages of more mature DP subsets were observed in *Nir3*$^{-/-}$ mice compared with WT mice (Fig. 4a).

To establish whether Nir3 regulates positive selection in a T cell-intrinsic manner, we performed a competitive bone marrow (BM) repopulation study. Equal numbers of BM cells from WT CD45.1/CD45.2 heterozygous mice and from *Nir3*$^{-/-}$ CD45.2 mice were adoptively transferred into lethally irradiated CD45.1 recipients. Six weeks after the cell transfer,

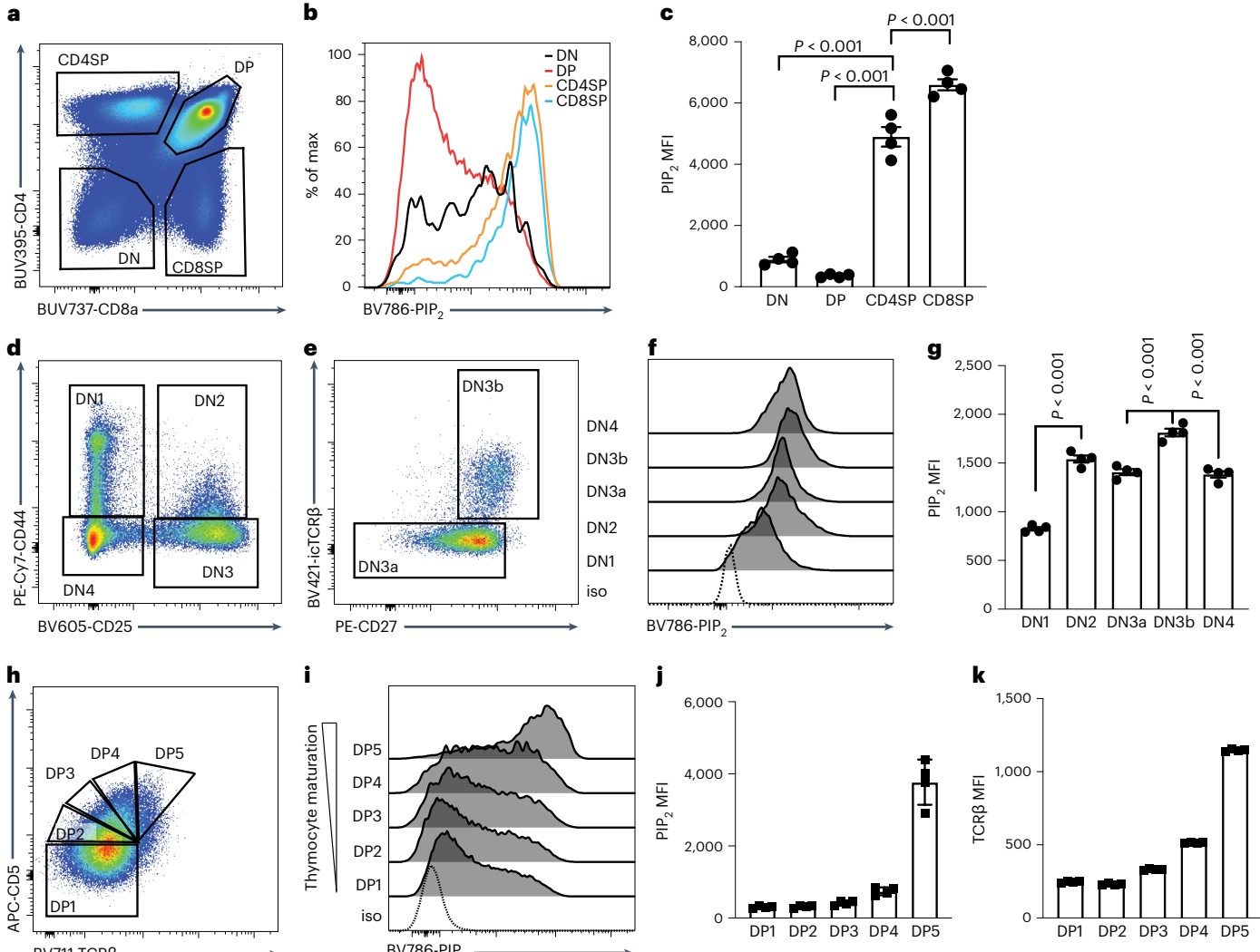

**Fig. 2 | PIP$_2$ is regulated through thymocyte development. a**, Total thymocytes from 6- to 8-week-old C57BL/6 female mice were divided into DN, DP, CD4SP and CD8SP populations based on expression of CD4 and CD8α (CD8). **b,c**, Histograms (b) and quantification (c) of PIP$_2$ in each thymocyte subset are depicted ($n = 4$) MFI, mean fluorescence intensity. **d**, DN cells were divided into four stages (DN1–DN4) based on the expression of CD44 and CD25. **e**, DN3 cells were subgated into DN3a and DN3b subsets based on intracellular TCRβ (icTCRβ) and surface expression of CD27 assessed by flow cytometry. **f,g**, Histograms (f)

and quantification (g) of PIP$_2$ in each DN subset were assessed. **h**, DP thymocytes were subdivided based on amounts of CD5 and TCRβ (DP1 = CD5$^{lo}$TCRβ$^{lo}$, DP2 = CD5$^{int}$TCRβ$^{lo}$, DP3 = CD5$^{hi}$TCRβ$^{lo}$, DP4 = CD5$^{hi}$TCRβ$^{int}$, DP5 = CD5$^{hi}$TCRβ$^{hi}$). **i,j**, Histograms (i) and quantification (j) of PIP$_2$ in each DP subset were assessed and are presented. iso, isotype control. **k**, Quantification of TCRβ in each DP subset. $n = 4$. Data are shown as the mean ± s.e.m. and are representative of three independent experiments. $P$ values were determined using an unpaired, one-way analysis of variance (ANOVA) with multiple comparisons.

repopulated thymi were analyzed and WT and *Nir3*$^{-/-}$ cells were identified using CD45.1 and CD45.2 congenic markers (Fig. 4b). There were roughly equal numbers of WT and *Nir3*$^{-/-}$ cells in the DP1–DP4 stages, which mirrored the 1:1 donor cell ratio. At DP5, CD4SP and CD8SP stages, the percentages of WT cells were greater than *Nir3*$^{-/-}$ cells, which suggests fewer *Nir3*$^{-/-}$ cells passed the positive selection checkpoint and transited into the DP5 stage and SP stages (Fig. 4c). These results are consistent with Nir3 playing an important role in positive selection, which occurs at the transition from DP4 to DP5 stages of thymocyte development.

It is common to find that positive selection of the normal diverse TCR repertoire is plastic enough to compensate at least partially for alterations in TCR signal strength by a repertoire shift[20]. TCR repertoires in *Nir3*$^{-/-}$ CD4SP and CD8SP were assessed by sampling selected V-J rearrangements of TCRα via real-time PCR. *Trav13* and *Trav16* segments were selected for their favored pairing with proximal Jα segments or equally usage proximal and distal Jα segments, respectively[21].

In CD4SP thymocytes, rearrangements of both Vα segments to Jα segments were comparable between WT and *Nir3*$^{-/-}$ cells. Yet, in *Nir3*$^{-/-}$ CD8SP thymocytes, rearrangements of both Vα segments to proximal Jα segments were markedly increased, whereas rearrangements of *Trav13* to the most distal Jα2 segment was markedly reduced (Extended Data Fig. 2). Our data indicate biased selection of a different TCR repertoire in *Nir3*$^{-/-}$ mice, at least in the CD8 lineage.

To avoid the impact of repertoire shifts in diverse TCR repertoire mice, we analyzed the effect of *Nir3* deficiency on thymocyte development in OT-I TCR transgenic mice (OVA pMHC reactive TCR), in which all T cells express the same transgenic TCR. We firstly noticed an increase in total thymic cellularity in *Nir3*$^{-/-}$ OT-I mice compared to WT OT-I mice (Fig. 4d). The defect in positive selection was much more pronounced in *Nir3*$^{-/-}$ OT-I mice than in *Nir3*$^{-/-}$ mice with a diverse TCR repertoire. Both the percentages and absolute numbers of CD8SP thymocytes in *Nir3*$^{-/-}$ OT-I mice were substantially lower compared with WT OT-I mice (Fig. 4e–g). Within the DP thymocytes, we observed pronounced

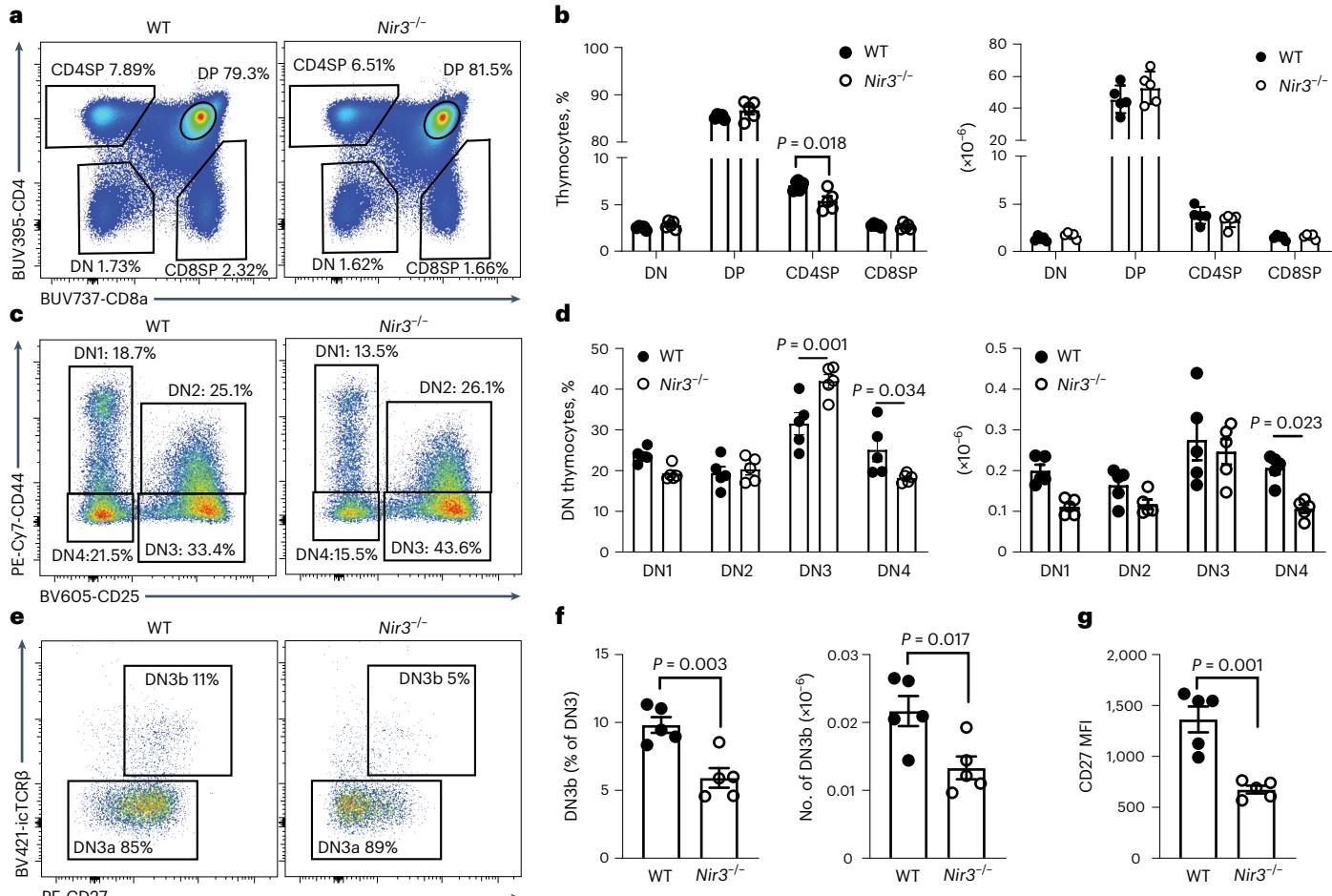

**Fig. 3 | *Nir3* deficiency leads to a blockade of thymocyte development at the TCRβ-selection stage. a,b,** Representative flow plots (a) and quantification (b) of CD4 versus CD8 profiles of thymocytes from 8-week-old *Nir3*[−/−] and WT female mice. **c,d,** Representative flow plots (c) and quantification (d) of DN1 to DN4 subsets of DN thymocytes. **e,f,** Representative flow plots (e) and quantification (f) of DN3a and DN3b subsets of DN3 thymocytes. **g,** CD27 expression on DN3 cells. *n* = 5. Data are representative of at least three independent experiments and are expressed as mean the ± s.e.m. *P* values were determined using two-way ANOVA with multiple comparisons (b,d) or an unpaired, two-tailed Student's *t*-test (f,g).

increased DP1 and decreased DP5 cells in *Nir3*[−/−] OT-I mice compared to WT OT-I mice (Fig. 4h,i). Fixing the TCR specificity in OT-I mice better highlighted the important role for Nir3 in thymocyte development.

**Role of Nir3 in agonist-selected T cell development in thymus**

During development, DP thymocytes expressing self-reactive TCRs are either deleted via negative selection or diverted into alternative T cell lineages, maintaining their self-reactivity. Lineage diversion of such self-reactive thymocytes is known as agonist selection and gives rise to natural killer T (NKT) cells, thymic-derived regulatory T (T$_{reg}$ cells) and intraepithelial lymphocyte precursors (IELp's)[22]. Although the environmental factors and signaling cues specifying these divergent fates remain unclear, one notion posits that intermediate TCR self-reactivities between positive and negative selection are required for agonist selection. However, recent studies have challenged this theory. TCR repertoire analysis suggested that T$_{reg}$ TCR repertoire may be quite broad overlapping with the conventional CD4 T cells and thymocytes undergoing negative selection[23]. In mice expressing a Nur77-GFP reporter, the expression of which correlates with the strength of TCR signaling, Nur77-GFP expression in thymic T$_{reg}$ and NKT cells is broad and largely overlapping with conventional CD4SP thymocytes[24]. On the other hand, studies using mice expressing OT-I TCR along with transgenic expression of the agonist ligand by medullary thymic epithelial cells (RIPmOVA mice) revealed the affinity threshold for thymic IELp development was higher than for negative selection[25]. Collectively, these findings suggest that the selection thresholds of TCR signaling strength in some T$_{reg}$ and NKT cells can be as low as the threshold for selection of conventional T cells, whereas the selection thresholds of IELp's are much higher. Consequently, based on our observed role of Nir3 in positive selection of conventional T cells, we hypothesized that Nir3 may also regulate development of T$_{reg}$ and NKT cells, but not IELp's.

Among these agonist-selected T cells, *Nir3* transcripts are highly expressed in thymic NKT cells (Fig. 1b). NKT cells are a subset of T cells expressing CD1d-restricted TCRs[26]. NKT cells can be subdivided into NKT1, NKT2 and NKT17 cells based on different patterns of cytokine secretion[27]. In *Nir3*[−/−] mice, the percentages of NKT cells in CD8-thymocytes and the number of NKT cells in the thymus were lower compared with WT mice (Extended Data Fig. 3a,b). *Nir3*[−/−] NKT cells express significantly less PLZF (Extended Data Fig. 3c,d), a critical transcription factor for NKT differentiation induced by NFAT signaling[28]. Based on the expression levels of transcription factor PLZF and RORγt, *Nir3*[−/−] mice developed significantly fewer NKT1 and NKT2 cells but comparable numbers of NKT17 cells compared with WT mice (Extended Data Fig. 3e–g).

We studied the thymic T$_{reg}$ cell development in *Nir3*[−/−] mice and in BM chimeric mice. In both cases, *Nir3*[−/−] mice or cells develop 30%-50% fewer natural T$_{reg}$ cells in the thymus (Extended Data Fig. 4). In contrast, we did not observe significant changes in the frequency of

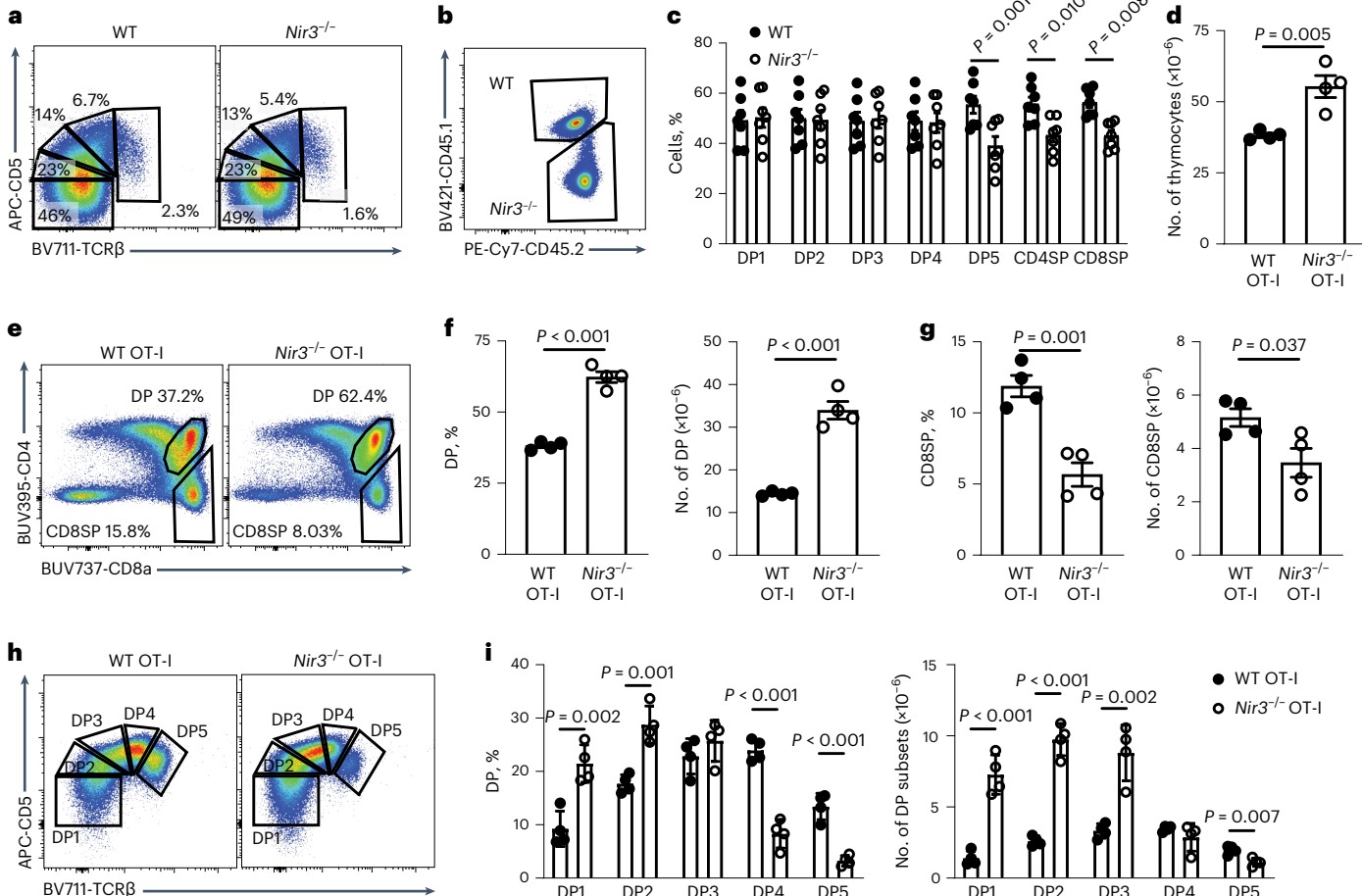

**Fig. 4 | Positive selection is modulated by Nir3. a**, Positive selection was visualized by TCRβ and CD5 in *Nir3*⁻/⁻ and WT DP thymocytes. **b,c**, BM cells from 6- to 8-week-old CD45.1/CD45.2 *Nir3* sufficient female mice and CD45.2 *Nir3*⁻/⁻ female mice were mixed at 1:1 ratio and adoptively transferred into 8-week-old CD45.1 female hosts. Six weeks after the transfer, WT and *Nir3*⁻/⁻ thymocytes at each development stage were identified by congenic CD45.1 and CD45.2 markers (b). The percentages of WT cells and *Nir3*⁻/⁻ cells in each stage were quantified (c). **d**, Number of total thymocytes in WT and *Nir3*⁻/⁻ OT-I mice were quantified. **e–g**, Representative flow plots (e) and quantification (f,g) of CD4 versus CD8 profiles of thymocytes from 8-week-old *Nir3*⁻/⁻ OT-I and WT OT-I female mice are shown. **h,i**, Positive selection was visualized by TCRβ and CD5 expression on *Nir3*⁻/⁻ OT-I and WT OT-I DP thymocytes (h) and quantified (i). *n* = 4 mice. Data are shown as the mean ± s.e.m. and are representative of two or three independent experiments. *P* values were determined using an unpaired, two-tailed Student's *t*-test.

IELp's generated from WT and *Nir3*⁻/⁻ cells in the BM chimeric mice (Extended Data Fig. 5). Overall, our data revealed the differential requirements for Nir3 during development of different subsets of agonist-selected T cells.

**Lymphopenia in *Nir3*⁻/⁻ peripheral lymphoid organs**
Compared to WT mice, *Nir3*⁻/⁻ mice had significantly fewer cells in the spleen and inguinal lymph nodes, but a comparable number of cells in the mesenteric lymph node harvested from unimmunized 8-week-old mice (Fig. 5a–c). Fewer CD4 and CD8 T cells were identified in the spleens of *Nir3*⁻/⁻ mice than in the spleens of WT mice (Fig. 5d). To determine whether the *Nir3* deficiency led to lymphopenia in a T cell-intrinsic manner, we performed a competitive BM repopulation experiment. Six weeks after BM cell transfer, DN thymocytes of the recipient mice consisted of roughly 50% WT and 50% *Nir3*⁻/⁻ cells, which reflected the initial 1:1 ratio of transferred BM cells. In the spleens of the recipient mice, there were significantly more WT T cells than *Nir3*⁻/⁻ cells in both CD4 and CD8 compartments (Fig. 5e). In *Nir3*⁻/⁻ CD4 cells, there were lower percentages of effector memory-like (TEM) cells compared to WT CD4 cells. *Nir3*⁻/⁻ CD8 lineage T cells were composed of fewer TEM cells and central memory-like (TCM) cells compared to WT CD8 cells (Fig. 5f,g). Moreover, we studied the BM chimeras with OT-II TCR transgenics, which showed similar phenotypes compared

with the polyclonal repertoire BM study. *Nir3*⁻/⁻ OT-II BM cells generated similar numbers of preselection thymocytes, but fewer numbers of postselection thymocytes and peripheral T cells (Extended Data Fig. 6). In sum, our data revealed that *Nir3* deficiency led to lymphopenia and fewer memory-like T cells in peripheral lymphoid organs.

**Nir3 regulates mature T cell survival in the periphery**
The positive selection blockade resulting from *Nir3* deficiency may contribute to lymphopenia in the secondary lymph organs. On the other hand, a reduction in CD44ʰⁱ *Nir3*⁻/⁻ cells potentially reflects less homeostatic T cell expansion in the periphery[29]. To study the role of Nir3 in T cell survival in the periphery, CellTrace dye-labeled WT and *Nir3*⁻/⁻ mature T cells were mixed and adoptively transferred into lymphoreplete WT recipient mice. Four and 10 days after the transfer, the percentages of WT CD4 and CD8 T cells were increased, suggesting a survival advantage of WT over *Nir3*⁻/⁻ T cells (Fig. 6a). Moreover, *Nir3*⁻/⁻ OT-I and OT-II cells showed more severe survival disadvantages in naive recipient mice compared with WT OT-I and OT-II cells, respectively (Fig. 6b). In lymphoreplete recipient mice, most of the adoptively transferred T cells had not undergone cell division, reflected by their largely undiluted CellTrace dye (Fig. 6c,d). Significantly fewer transferred *Nir3*⁻/⁻ OT-I and OT-II cells expressed Ki67, indicating cell cycle entry, compared with WT OT-I and OT-II cells in naive recipient

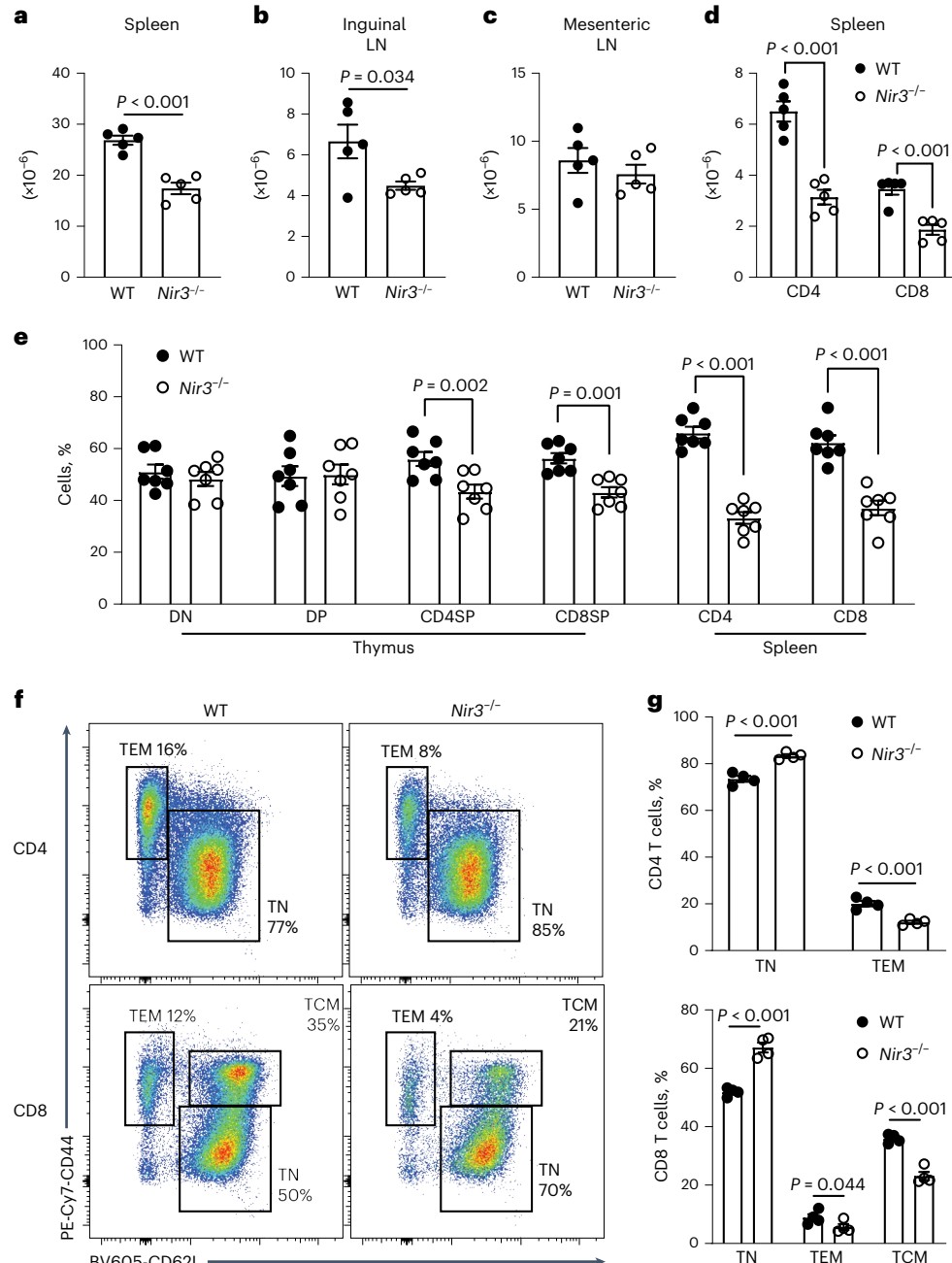

**Fig. 5 | Lymphopenia in *Nir3*−/− mice. a–c**, Total cell number in spleens (a), inguinal lymph nodes (b) and mesenteric lymph nodes (c) from WT and *Nir3*−/− female mice. **d**, Numbers of CD4 and CD8 cells in spleens from 8-week-old WT and *Nir3*−/− female mice. *n* = 5 mice. **e–g**, BM cells from 6- to 8-week-old CD45.1/CD45.2 *Nir3* sufficient female mice and CD45.2 *Nir3*−/− female mice were mixed at a 1:1 ratio and adoptively transferred into 8-week-old CD45.1 female hosts for 6 weeks. **e**, The percentages of WT and *Nir3*−/− cells in each subset. DN, DN thymocytes; DP, DP thymocytes; CD4SP, CD4SP thymocytes; CD8SP, CD8SP thymocytes; CD4, splenic CD4 T cells; CD8, splenic CD8 T cells. *n* = 8 mice. **f,g**, WT and *Nir3*−/− splenic CD4 and CD8 T cells were subdivided into naive and memory-like subsets based on their expression of CD44 and CD62L. **f,g**, Representative flow plots (f) and quantification (g) of percentages of each subset. *n* = 4 mice. Data are shown as the mean ± s.e.m. and are representative of at least two independent experiments. *P* values were determined using an unpaired, two-tailed Student's *t*-test.

mice, respectively (Fig. 6c–e). We further examined the homeostatic proliferative potentials of *Nir3*−/− T cells in lymphopenic *Tcra*−/− mice. *Nir3*−/− OT-I and OT-II cells expressed less Ki67 compared with WT OT-I and OT-II cells (Fig. 6f–h). Our data suggest that Nir3 plays an important role in survival of mature T cells in the periphery, possibly by regulating TCR signaling in response to self-pMHC.

## Nir3 promotes PIP₂ replenishment following TCR stimulation

The defects in T cell development and survival in the periphery suggested a role for Nir3 in response to weak TCR stimuli. We hypothesized that Nir3 regulated the replenishment of PM $PIP_2$ levels in T cells following TCR stimulation. *Nir3*−/− DP thymocytes and mature T cells have similar basal $PIP_2$ levels with WT DP thymocytes and mature T cells, respectively (Extended Data Fig. 7). *Nir3*−/− T lineage cells replenished $PIP_2$ levels significantly more slowly after soluble anti-CD3 antibody stimulation (Fig. 7a). The defect is more pronounced in DP thymocytes and CD4 T cells than CD8 T cells, possibly due to relatively high Nir3 expression in thymocytes and relatively low expression in mature CD8 cells (Fig. 1d). Because Nir2 is more abundant than Nir3 in peripheral T cells (Fig. 1d), we compared Nir2 with Nir3 for their contributions to

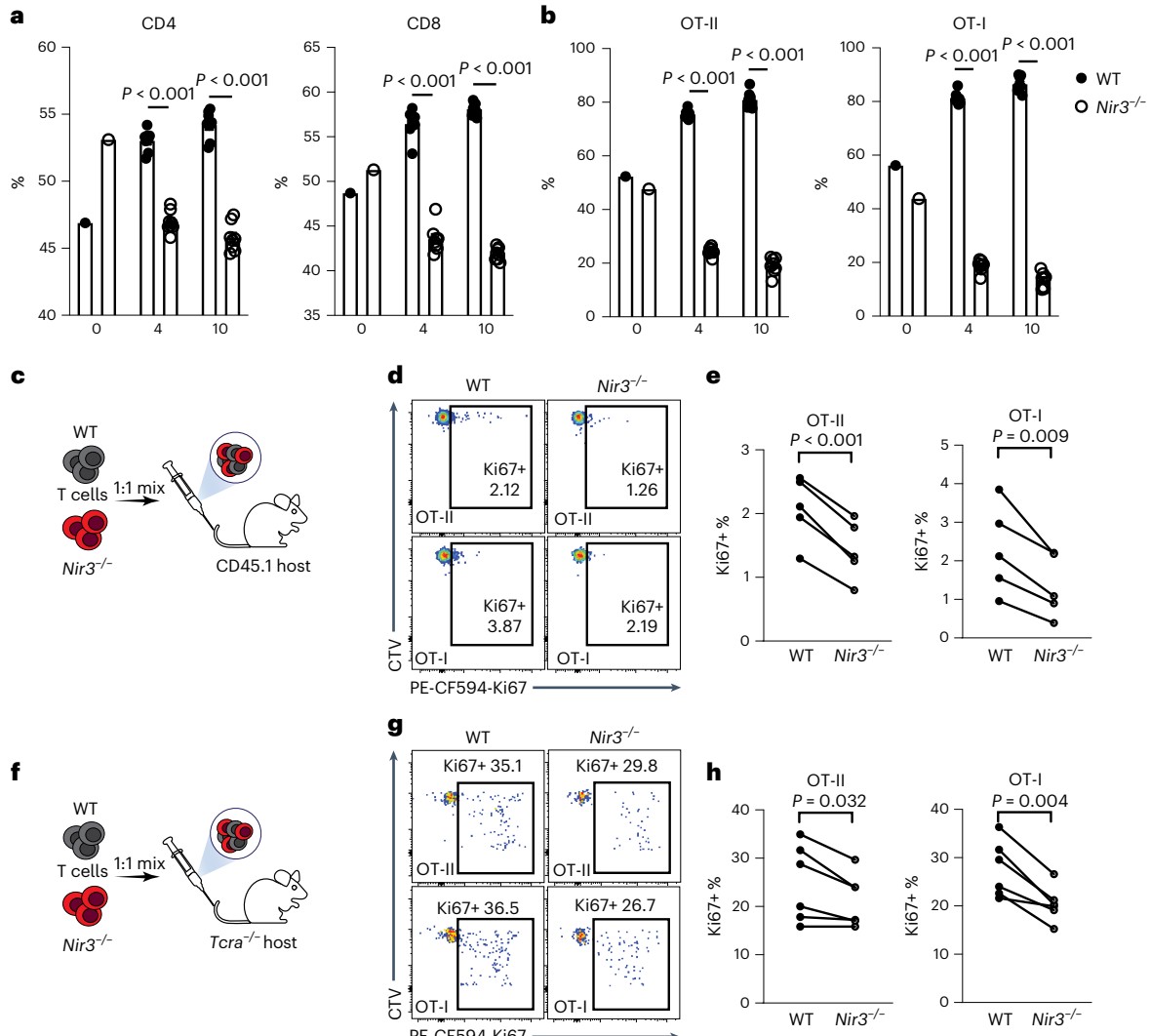

**Fig. 6 | Nir3 regulates mature T cell survival in the periphery. a,b**, WT T cells and *Nir3*[−/−] T cells labeled by CellTrace dyes were mixed at a 1:1 ratio and adoptively transferred into 6- to 8-week-old WT male recipient mice. Four and 10 days after the transfer, the percentages of transferred WT CD4, CD8 (a), OT-II, or OT-I (b) in total surviving transferred cells were quantified. *n* = 8 mice. **c–h**, WT and *Nir3*[−/−] OT-I and OT-II T cells labeled by CellTrace Violet dyes (CTV) were mixed at a 1:1 ratio and adoptively transferred into CD45.1 male recipient mice (c–e) or *Tcra*[−/−] male mice (f–h) for 4 days. Representative flow plots (d,g) and quantification (e,h) of Ki67-expressing cells in transferred cells are shown. WT and *Nir3*[−/−] donor cells recovered from the same recipient mouse were connected with lines. *n* = 5 mice. Data are shown as the mean ± s.e.m. and are representative of at least two independent experiments. *P* values were determined using an unpaired (a,b) or a paired (e,h), two-tailed Student's *t*-test.

PIP$_2$ replenishment by delivering Nir2 siRNAs to WT and *Nir3*[−/−] T cell blasts (Extended Data Fig. 8a). PIP$_2$ replenishment following TCR stimulation was more impaired in Nir2 knockdown cells than *Nir3*[−/−] cells, whereas PIP$_2$ replenishment was most severely attenuated in Nir2 knockdown in *Nir3*[−/−] cells (Extended Data Fig. 8b).

We further tested the role of Nir3 in PIP$_2$ replenishment in the PMs of mature T cells in response to immobilized anti-CD3 antibody stimulation. We stimulated Tubby-mScarlet (a PIP$_2$ biosensor) expressing *Nir3*[−/−] OT-II cells on coverslips coated with anti-CD3 antibody. PIP$_2$ levels were quantified by the MFI of the Tubby-mScarlet biosensor in the PMs of spreading cells on glass using total internal reflection fluorescence (TIRF) microscopy. The PIP$_2$ levels in *Nir3*[−/−] OT-II cells were more strikingly depleted and not replenished compared with WT OT-II cells (Fig. 7b,c). Because anti-CD3 monoclonal antibody is such a potent stimulus, we stimulated the Tubby-mScarlet expressing *Nir3*[−/−] OT-I cells on coverslips coated with full (OVA), partial (T4) or weak agonist (G4) OVA pMHC tetramers. The depletion of PIP$_2$ was more sustained and the replenishment of PIP$_2$ levels was much slower

in *Nir3*[−/−] OT-I cells than in WT OT-I cells in all three different stimulation conditions (Fig. 7d–i). During the PIP$_2$ replenishment, we noticed that Tubby formed punctate patterns on the PMs, which suggested PIP$_2$ synthesis is clustered, probably at sites of PI delivery at ER-PM junctions (Fig. 7b,d,f,h). Our data largely support our hypothesis that Nir3 plays an important role of PIP$_2$ replenishment in thymocytes and mature T cells in response to TCR stimulation.

## Nir3 regulates TCR downstream signaling in DP thymocytes
We assessed whether the defects in PIP$_2$ replenishment led to impaired of TCR signaling in *Nir3*[−/−] thymocytes. In WT DP thymocytes, consistent with our previous report, peak TCR-induced [Ca$^{2+}$]$_i$ elevation increases from DP1 to DP5 stages of maturation (Fig. 8)[17]. In *Nir3*[−/−] DP5 thymocytes, stimulation with a lower concentration of anti-CD3 antibodies (0.1μg/ml) led to more impaired [Ca$^{2+}$]$_i$ elevation compared with WT DP5 thymocytes. Compared with the most mature DP5 cells, *Nir3* deficiency has a greater impact on TCR-induced [Ca$^{2+}$]$_i$ elevation in less mature DP1-DP4 thymocytes, which also express fewer cell surface

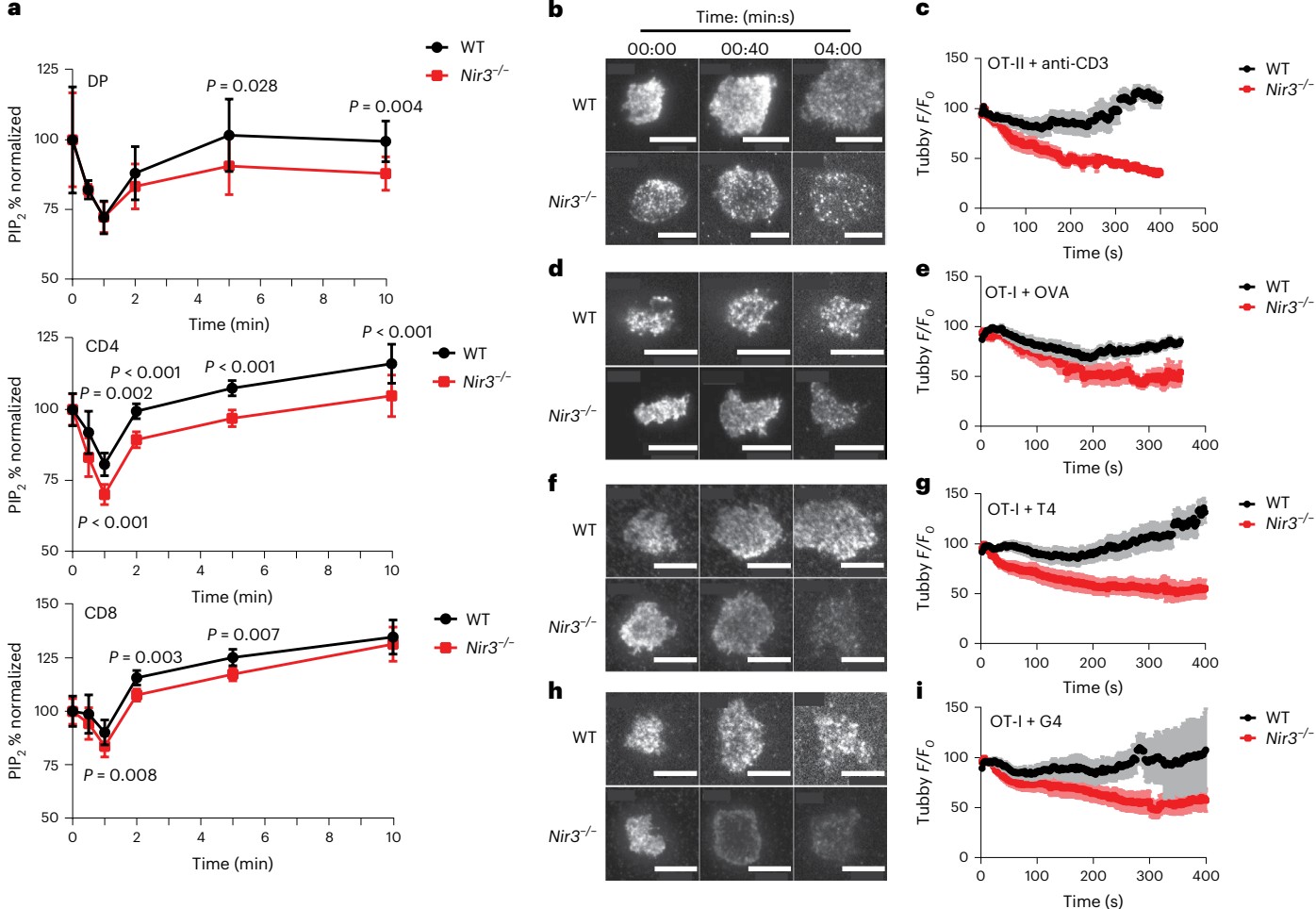

**Fig. 7 | Nir3 contributes to rapid PIP$_2$ replenishment following receptor-induced hydrolysis in thymocytes. a**, Thymocytes and mature T cells from WT and *Nir3*$^{-/-}$ male mice were stimulated with anti-CD3 for the indicated times. PIP$_2$ levels were normalized to unstimulated cells (t = 0). $n$ = 4. **b,c**, OT-II CD4 T cells expressing Tubby-mScarlet were stimulated on coverslips coated with anti-CD3 antibody. Representative images (b) and quantification (c) of Tubby MFI on the spreading membrane are shown. **d–i**, OT-I CD8 T cells expressing Tubby-mScarlet were stimulated on coverslips coated with full (OVA), partial (T4) or weak (G4) agonist OVA peptide-MHC tetramers. Representative images (d,f,h) and quantification (e,g,i) of Tubby MFI on the spreading membrane were shown. Scale bar=10µm. N > 8 biological independent cells. All data are shown as the mean ± s.e.m. and are representative of at least two independent experiments. *P* values were determined using a paired, two-tailed Student's *t*-test.

TCR molecules and lower PIP$_2$ levels (Fig. 8). Of note, stimulation of the thymocytes at various stages of DP development with thapsigargin or ionomycin had no differential effect since these reagents increase [Ca$^{2+}$]$_i$ via mechanisms that bypass the PIP$_2$ dependent events at the PM. Overall, our results suggested that Nir3 is critical for [Ca$^{2+}$]$_i$ elevation induced by weaker TCR stimulation in thymocytes.

We next assessed whether Nir3 regulates TCR-induced Erk phosphorylation (pErk) as a consequence of PIP$_2$ hydrolysis and the generation of DAG and its regulation of the RasGRP guanine nucleotide exchange factor[30]. Upon stimulation with 10 µg ml$^{-1}$ anti-CD3 antibodies, *Nir3*$^{-/-}$ DP thymocytes at different maturation stages exhibited similar levels of pErk compared with WT thymocytes. However, upon stimulation with 1 µg ml$^{-1}$ anti-CD3 antibodies, *Nir3*$^{-/-}$ DP thymocytes at DP4 and DP5 stages exhibited substantially less pErk compared with WT thymocytes at the same stages (Extended Data Fig. 9a). This suggests Nir3 has a greater impact in regulating the activation of the Ras/MAPK pathway in response to weak TCR stimuli rather than strong TCR stimuli in DP thymocytes.

Next, we tested the activation thresholds of *Nir3*$^{-/-}$ DP1 and DP5 thymocytes in response to agonist pMHC ligands of different potencies. Nur77 expression was used as a readout of TCR signaling strength, and

CD69 upregulation was selected as an integrated activation marker[31]. *Nir3*$^{-/-}$ DP1 thymocytes expressed substantially less Nur77 than WT DP1 thymocytes at the basal level and after stimulation with strong (OVA) or weak peptide (G4). On the other hand, *Nir3*$^{-/-}$ DP5 thymocytes expressed comparable levels of Nur77 compared with WT DP5 thymocytes in response to full agonist peptide (OVA) stimulation, but slightly less Nur77 with positive selecting peptide Cappa1 stimulation (Extended Data Fig. 9b).

Our data demonstrate that Nir3 is involved in regulating TCR signaling in DP thymocytes and likely in both pre-TCR signaling during TCRβ-selection and in peripheral T cell tonic signaling. The accumulation of TCR signaling defects was observed at different stages of *Nir3*$^{-/-}$ thymocyte development and is likely to contribute to T cell survival defects in *Nir3*$^{-/-}$ mice. Our data also suggests that differential expression of Nir2 and Nir3, together with other calcium signaling molecules, may contribute to the TCR sensitivity difference between preselection thymocytes and mature T cells (Extended Data Fig. 10).

## Discussion

Among five PITPs in the human genome, only Nir2 and Nir3 have been shown to dynamically localize at ER-PM junctions following

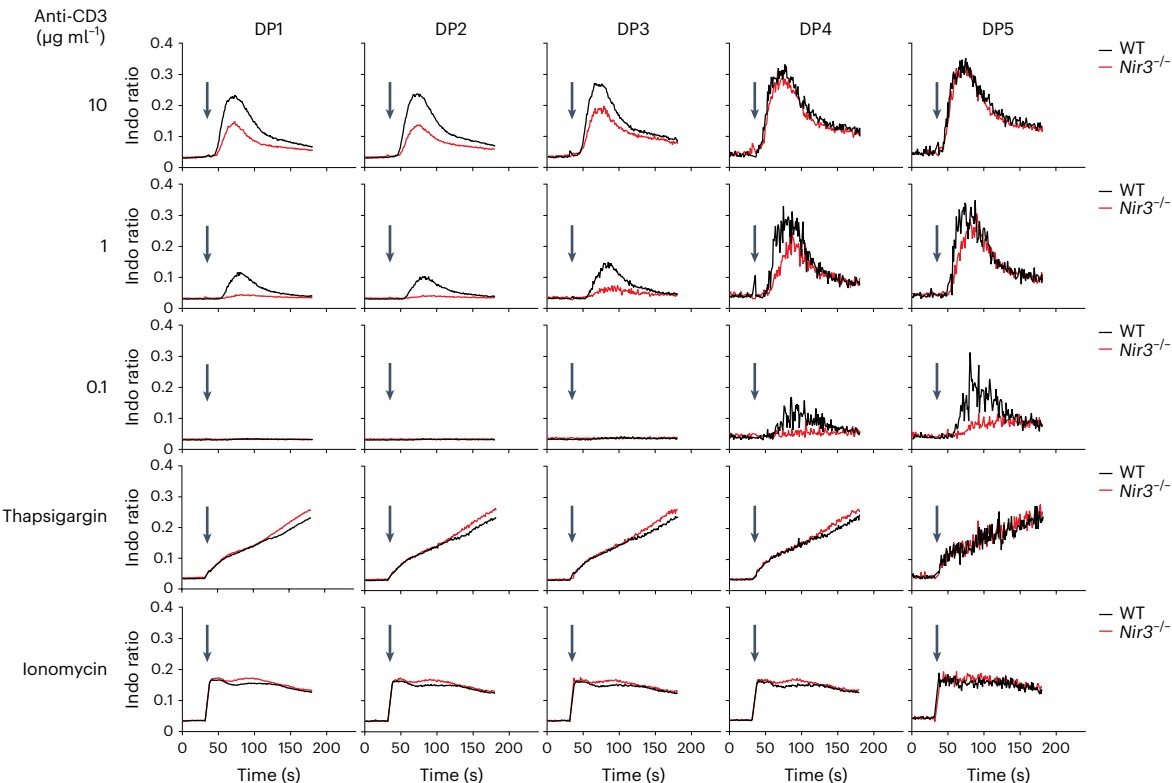

**Fig. 8 | Nir3 regulates TCR-induced $[Ca^{2+}]_i$ in DP thymocytes.** WT or $Nir3^{-/-}$ DP thymocytes loaded with the calcium sensitive dye Indo-1 AM were incubated with indicated amount of anti-CD3 antibody at time 0 and stimulated with crosslinking antibodies, thapsigargin or ionomycin at the arrow. The $[Ca^{2+}]_i$ elevation of each DP subset, as gated in Fig. 4a, was assessed based on the Indo-1 fluorescence emission ratio 405nm/485nm by flow cytometry. $n > 5$ cells. All data are shown as the mean ± s.e.m. and are representative of at least two independent experiments.

receptor-induced $PIP_2$ hydrolysis and to regulate $PIP_2$ levels[32]. In this study, we report that Nir3, the expression of which is preferentially upregulated in preselection thymocytes, plays an important role in thymocyte development and in mature T cell survival in the periphery. Mechanistically, by transporting the precursor PI from the ER to the PM Nir3 maintains and replenishes PM $PIP_2$ levels in response to TCR simulation by weak ligands. Rapid replenishment of PM $PIP_2$ levels supports adequate TCR-induced calcium signaling involved in maturation, developmental checkpoints and survival.

In unstimulated thymocytes and T cells, $Nir3$ deficiency did not affect basal $PIP_2$ levels. Besides nonvesicular lipid transport which is mediated by PITPs at ER-PM junctions, the ER exchanges bulk lipids with the PM through vesicular lipid transport via the Golgi apparatus[33]. It is more plausible that the amount of PI for maintaining basal $PIP_2$ levels is largely transported through the vesicular transport in quiescent T cells. Compared with the slow lipid transport by vesicular traffic, which could take one hour or longer, nonvesilcular lipid transport at the ER-PM junctions is much faster and occurs over time frames of seconds or a few minutes[33–35]. In T cells, only such a rapid transfer mechanism might be required compensate for the swift $PIP_2$ depletion in the PM induced by TCR stimulation, which makes ER-PM junctions potentially important subcellular structures for TCR-induced calcium signaling.

Following TCR stimulation, new ER-PM junctions form rapidly following the ER calcium depletion[36]. In addition to transferring PI, ER-PM junctions could provide docking sites for various TCR signaling molecules. For example, scaffolding proteins such as Tespa1 and SLAT could potentially contribute to the recruitment of PLCγ1 and $IP_3$ receptors ($IP_3$Rs) to the same macromolecular LAT signaling complex at the ER-PM junctions. This could increase signal transduction rates from $PIP_2$ hydrolysis to ER calcium depletion[37,38]. Most ER-PM junctions

are formed in gaps within the cortical actin mesh structure[39]. Our previous studies utilizing a model Csk inhibition system, have suggested that cortical actin cytoskeleton may act as a barrier to TCR signaling by sequestering activated LAT-bound PLCγ1 from its substrate $PIP_2$ on the PM[40]. It is possible that TCR stimulation contributes to the formation of new or larger gaps in the cortical actin cytoskeleton via the activation of Rac/Rho GEFs and Rac/Rho GTPases[41,42]. Such remodeling could permit new and expanded ER-PM junctions that could promote the access of PLCγ1 to $PIP_2$ and the consequent hydrolysis of $PIP_2$. Hydrolysis of $PIP_2$ and generation of DAG triggers the release of PIP5 kinases from the PM which could further enhance the actin clearing at the immune synapses[43]. Taken together, ER-PM junctions are potentially critical subcellular structures in T cells for calcium-mediated signal transduction. Our findings highlight the importance of Nir3 in TCR-induced calcium signaling at ER-PM junctions.

Proper TCR-induced calcium signaling is important for both TCRβ selection and positive selection in thymus. Mice carrying a point mutation in the PLCγ1 binding site of LAT (Y136F) exhibited a severe block of thymocyte development at the DN3 stage before TCRβ selection[44,45]. In previous reports, deleting PLCγ1 at early DP stages led to a complete block of thymocyte development at the positive selection stage[46]. Mice lacking both Rlk and Itk, two Tec kinases phosphorylating PLCγ1 in T cells, showed defects at both TCRβ selection and positive selection steps[47]. In this study, we found that $PIP_2$ levels in thymocytes increase significantly as cells pass both TCRβ selection and positive selection. The increase of $PIP_2$ is concordant with both basal and TCR-induced $[Ca^{2+}]_i$ as thymocytes mature[17]. The increase of $PIP_2$ can influence the TCR threshold and facilitate thymocyte passage through these selection checkpoints. Indeed, we found that thymocytes are more sensitive to weaker TCR stimulation as they approach the positive selection checkpoints. Likewise, the signaling via the pre-TCR is thought to

deliver weak signaling as well. Nir3 contributes greatly to the increased sensitivity to weaker TCR stimulation in immature thymocytes.

The functional differences between Nir2 and Nir3 are mostly quantitative rather than qualitative. Nir3 has a reduced rate of PI transfer but an enhanced ability to sense PA on the PM[10]. These features enable Nir3 to rapidly replenish PM $PIP_2$ levels in response to weak TCR stimulation. With its higher expression in preselection thymocytes, Nir3 appears to be an elaborately evolved gene specialized for T cell development. Such a highly specialized gene that is well suited for T cell development is not an exception and can be repeatedly found in mammalian genomes. The $IP_3$ receptor type 2 (ITPR2) and stromal interaction molecule 2 (STIM2) are two other examples which are also involved in TCR-induced calcium signaling. $IP_3$ receptors sense $IP_3$ generated by $PIP_2$ hydrolysis and release calcium from the ER lumen to the cytosol. There are three $IP_3$ receptor subunits in the mammalian genome, ITPR1, ITPR2 and ITPR3. ITPR2 has the highest affinity to $IP_3$ and highest expression level in preselection thymocytes, whereas ITPR3 has the least affinity to $IP_3$ and highest expression level in mature T cells[48]. Moreover, STIM1 and STIM2 are the molecular sensors of the ER calcium depletion. They oligomerize upon ER calcium depletion and open the Orai calcium channels on the PM to induce store-operated calcium entry. Compared with STIM1, STIM2 is more sensitive to the depletion of ER calcium. Expression profiles of the two STIM genes during the T cell development revealed that expression of STIM2 peaks at preselection thymocytes, whereas STIM1 is dominantly expressed in mature T cells[49]. Like Nir2 and Nir3, the separation of $IP_3$ receptor genes and STIM genes took place at the origin of vertebrates[49,50]. After 500 million years of evolution, the thymocyte expressing paralog(s) of each gene have all evolved with enhanced sensitivity for events involved in TCR signaling compared with their paralogs expressed in mature T cells. These findings suggest that an extremely sensitive calcium signaling machinery in response to weak TCR stimulation is an evolutionary adaptation for thymocyte development.

## Online content

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

## Methods

### Mice

To generate *Nir3*⁻/⁻ mice, a *Nir3*-targeting guide RNA that targets Cas9 to a site near in the fourth exon of *Nir3* and purified Cas9 were obtained from PNA Bio INC. Purified Cas9 and *Nir3*-targeting gRNA were injected into C57BL/6 zygotes at the UCSF Diabetes Research Center Core. The resulting pups carrying a 98 bp deletion were genotyped by amplifying a 413 bp region flanking the targeted region (Supplementary Fig. 1). The mice were backcrossed to C57BL/6 mice for at least six generations. BoyJ (CD45.1) mice were obtained from The Jackson Laboratory (B6.SJL-Ptprca Pepcb/BoyJ). CD45.1/CD45.2 mice were generated by crossing BoyJ and C57BL/6 mice. All mice used in these studies were housed in a specific-pathogen-free facility at UCSF according to the University Animal Care Committee and National Institutes of Health (NIH) guidelines.

### Antibodies and reagents

Antibodies and reagents are listed in the Supplementary Table 2.

### PIP₂ measurements

For measurements of $PIP_2$ in thymocytes using immunostaining, biotinylated monoclonal anti-$PIP_2$ antibody (clone 2C11, Echelon Biosciences) was used as previously described[51]. Briefly, thymocytes were stained for surface markers and rapidly fixed by 4% paraformaldehyde (PFA) and 0.2% glutaraldehyde (GA) for 15 min at 20 °C. Fixed cells were then placed ice for 2 min and all subsequent staining steps were performed on ice. Cells were blocked and permeabilized for 45 min with PIPES buffer (Thermo Scientific, pH 7.0) containing 5% (v/v) normal goat serum (NGS), 50 mM $NH_4Cl$ and 0.5% saponin. 2.5 µg ml⁻¹ anti-$PIP_2$ antibodies were applied in PIPES buffer with 5% NGS and 0.1% saponin for 1 hour. After two washes in PIPES buffer, a 45 min incubation with fluorescent conjugated streptavidin in PIPES buffer with 5% NGS and 0.1% saponin was performed. Cells were then rinsed four times with PIPES buffer and analyzed by flow cytometry.

For measurements of dynamic changes of $PIP_2$ at the PM during TCR stimulation, the intensity of $PIP_2$ biosensor Tubby was monitored by TIRF microscopy in OT-I and OT-II T cells stimulated in coverslip chambers. Glass surfaces of coverslip chambers were coated with either stimulatory antibody (anti-CD3) or streptavidin (SA). Surfaces were then washed and biotinylated monomeric pMHC complexes were incubated with SA surfaces. Surfaces were then washed and used for antigen presentation. OT-I or OT-II cells resuspended in the complete media were added in chambers coated with stimulatory reagents at 37 °C. Individual immunological synapses formed on stimulatory glasses were imaged with the Nikon Ti Microscope with TIRF every 15 s for at least 4 min after cells first interacting the glasses. Areas of immunological synapses were selected based on Tubby. Tubby MFI within each synapse in each frame was normalized to the first frame after cell interacting the glasses (time 00:00). The microscopy was controlled by the NIS-Elements AR imaging software v5.41.02 and a ×100 oil dipping objective was used. The data were processed by ImageJ software v2.3.0/1.53p.

### Competitive repopulations and chimeras

For competitive repopulation assays, BM harvested from the femurs and tibias from 8- to 12-week-old adults were injected intravenous into lethally irradiated (two doses of 493 rad) CD45.1 BoyJ hosts, and reconstituted thymi were analyzed 6–8 weeks later. At least $5 \times 10^6$ total BM cells were injected intravenous into each mouse. Within an experiment, all hosts were sex and age matched and were between 8 and 12 weeks old.

### Peripheral T cell isolation and transfection

Splenic T cells were isolated using Dynabead untouched negative selection kit (Stemcell EasySep) and stimulated with anti-CD3 and anti-CD28 antibody coated Dynabeads (Dynabeads Mouse T-Activator CD3/CD28, ThermoFisher) in RPMI 1640 medium supplemented with 10% FCS, Hepes, 50 µM 2-Mercaptoethanol and L-glutamine (T cell media). After 48 h, Dynabeads were removed. Cells were washed twice in PBS and resuspended at $5 \times 10^6$ cells per 100 µl in Neon transfection buffer R (Invitrogen) and mixed with 20 µM siRNA (Ambion/Life Technologies) before being immediately electroporated for $3 \times 10$ ms at 1,550 V on a Neon transfection instrument. The 100-µl transfection was then transferred directly into pre-warmed T cell media containing 10 ng ml⁻¹ IL-7 in a flat-bottomed 24-well plate (Costar) for another 48 h. Knockdown efficiencies were assessed by immunoblots. Scrambled siRNA: Silencer Select Negative Control No. 2 siRNA (ThermoFisher); Nir2 siRNA-1: 5′-GCUCUUGUCUCCAACCUGAtt-3′; Nir2 siRNA-2: 5′-AGAGAAUCGUGGAUACCAUtt-3′. To knockdown Nir2 in peripheral T cells, Nir2 siRNA-1 and siRNA-2 were equally mixed and transfected.

### OT-I co-culture

Splenocytes were used as antigen-presenting cells and were isolated from T cell-deficient mice (*Tcra*⁻/⁻ mice). Prior to culture, red blood cells were removed using ACK lysis. Splenocytes were incubated with peptide antigens for 1 h at 37 °C. OT-I thymocytes or T cell blasts were added at a ratio of 5:1 (APC/thymocytes). Cells were cultured 3 to 4 h at 37 °C and then stained for CD69 and other surface markers (CD8α, TCRβ or Vα2) using an antibody dilution of 1:200 in FACS buffer (1x PBS, 1% BSA, 0.5 mM EDTA, 2 mM $NaN_3$) and Fc blocking antibody (2.4G2) at 1:1,000. Washed cells were rapidly fixed by 4% paraformaldehyde for 15 min at 20 °C. Fixed cells were then permeabilized with 0.1% Triton X-100 (diluted in PBS) at 20 °C for 10 min. Anti-Nur77 antibodies were applied in FACS buffer for 30 min. After two washes in FACS buffer, upregulation of Nur77 and/or CD69 in cells were assessed by flow cytometry using a BD Fortessa and quantified using FlowJo software. The OVA peptide (SIINFEKL), its variants (T4, G4), the VSV peptide and the Cappa1 peptide (ISFKFDHL) were synthesized by Genescript.

### Retroviral transduction of mouse T cells

Naive OT-I or OT-II T cells were isolated using magnetic bead-mediated negative selection (Miltenyi Biotec). Isolated T cells were cultured with mouse T-Activator (anti-CD3/anti-CD28) dynabeads and 50 U ml⁻¹ IL-2 in a 24-well plate overnight. The retroviral supernatants were mixed with Lipofectamine, incubated at 25 °C for 20–30 min and then added into the T cell culture. The plate was wrapped in saran wrap and centrifuged at 460 *g* for 1 h at 25 °C. The plate was then moved to a 37 °C incubator. Expression of mScarlet was monitored using an LSR Fortessa (BD Biosciences) and samples were analyzed 24 h after transduction.

### Expression of *Nir2* and *Nir3*

The RNA expression levels of *Nir2* and *Nir3* across 55 human tissues were obtained from the Human Protein Atlas database (www.proteinatlas.org)[52]. The consensus dataset was used which consists of normalized expression levels for 55 tissue types, created by combining the HPA and GTEx transcriptomics datasets using their internal normalization pipeline. The RNA expression levels of *Nir2* and *Nir3* in mouse T cell subsets were determined from the ImmGen database (www.immgen.org)[53,54]. The ImmGen Microarray phase 1 dataset was used which consists of normalized expression levels for primary cells from 6-week-old C57BL/6 male mice.

The relative protein abundance of Nir2 and Nir3 in mouse thymocytes and mouse peripheral T cells was determined by immunoblots. Roughly equal amount of ectopically expressed Nir2-mCherry and Nir3-mCherry in HEK293 cells were loaded as normalization controls, which was confirmed by mCherry blots. The intensities of Nir2 and Nir3 protein bands of thymocytes or peripheral T cells were initially normalized by actin and subsequentially normalized by Nir2 or Nir3 bands of ectopically expressed Nir2-mcherry or Nir3-mcherry, respectively. The normalized amount of Nir2 and Nir3 in different cell subsets were further normalized to Nir2 levels in thymocytes, which were set to 1.

## Sequence alignments and analysis

Potential orthologues for Nir2 and Nir3 were derived from the Inter-Pro Scan database employing the following procedure. Using human protein sequences as inputs (UniProt ID's O00562 and Q9BZ72 for Nir2 and Nir3 respectively), a list of orthologues for each protein across species was obtained, stretching as far as sponges, a lineage at the base of animalia that diverged nearly ~1,000 million years ago from *Homo sapiens*. Subsequently, from each of these lists, sequences were filtered to remove sequences with outlier lengths (<750 or >1,750 amino acids), present originally due to paralogs of Nir1 as well as other isoforms of PITP-domain-containing proteins. When multiple orthologs were present for a single species, an annotated reference sequence was chosen if available, otherwise, the isoform with closest similarity to the human Nir2/Nir3 protein was retained. A joint sequence list was created, removing duplicates of orthologs in the same species and reflecting an underlying absence of one of the Nir2/Nir3 proteins.

MAFFT was used to construct a multiple sequence alignment for this list of sequences. TrimAI was used to trim for regions with more than 20% gaps and the trimmed multiple sequence alignment was used as input for phylogenetic reconstruction by maximum likelihood analyses through FastTree MP. The initial alignment covered over 750 sequences across 600 species and Phylogenetic Diversity Analyzer was used to generate a subsampled sequence list that simultaneously preserved tree features but with an emphasis on the vertebrate lineage with select but well-spaced outgroups including chordates, invertebrates and sponges (totally 200 sequences). This subsampled list was subsequently aligned, trimmed, and a tree was generated as described above, and annotated manually using the ITOL website.

## Phospho-Erk staining

Cells were stimulated in RPMI and fixed by adding BD Fixation/Permeabilization solution (1:1 ratio, PFA-based fixation reagent, BD Biosciences) and incubating for 10 min at 20 °C. Cells were pelleted and rinsed with FACS buffer (PBS supplemented with 2% FBS and 2 mM EDTA). Cells were then placed on ice and ice-cold 90% methanol was added to permeabilize the cells for 45 min. Cells were then rinsed three times with FACS buffer and resuspended in staining solution (anti-phospho-Erk 1:100 in FACS buffer). Cells were stained for either 1 h at 20 °C or overnight at 4 °C. Cells were rinsed three times and stained with anti-Rabbit PE antibody (1:100 in FACS buffer) for 30 min at 20 °C. Cells were rinsed twice and analyzed by flow cytometry using a LSR Fortessa controlled by BD FACSDiva v8.0.1 software and quantification performed using FlowJo software.

## Calcium assays

Cells were loaded with the Indo1-AM (1.5 µM, Invitrogen) for 30 min at 37 °C in RPMI with 10% fetal bovine serum, washed, surface stained and kept on ice in RPMI. Cells were warmed to 37 °C for 5 min before stimulation. Changes in Indo1 fluorescence in cells were recorded using a FlexStation or a LSR Fortessa controlled by BD FACSDiva v8.0.1 software. FACS data were exported from FlowJo in CSV format and analyzed with an R script (https://github.com/richard02050411/Calcium-Flux-FACs/blob/master/R-code). Graphs were generated using Graphpad Prism software.

## Quantification and statistical analysis

Statistical analysis was applied to biologically replicates for each experiment. No statistical methods were used to pre-determine sample sizes, but our sample sizes are similar to those reported in previous publications[55–57]. Data distribution was assumed to be normal but this was not formally tested. No randomization was used in experiments. In animal experiments, age matched animals were allocated based on their genotypes. In cell stimulation experiments, cells with the same genotype were pooled together and equally allocated into different groups before treatments. Data collection and analysis were not performed blind to the conditions of the experiment. The numbers of independent experiments performed with similar results are indicated in the figure legends. No animals or data points were excluded from the analyses in all experiments. All statistical analyses of experiments were performed using two-tailed Student *t*-tests, One-way or two-way ANOVA with multiple comparisons. GraphPad Prism 8 Software (GraphPad Software) was used for data analysis and representation. All bar graphs show means with overlaid scatter dots, or error bars (indicating s.e.m.), to show the distribution of the data, as indicated in each figure legend.

## Reporting summary

Further information on research design is available in the Nature Portfolio Reporting Summary linked to this article.

## Data availability

RNA expression profiles of *Nir2* and *Nir3* in human tissues and mouse T cell populations are available on the Human Protein Atlas (https://www.proteinatlas.org/) and Immunological Genome Project (https://www.immgen.org/), respectively. Source data are provided with this paper.

## Code availability

Increase of free cytoplasmic calcium was analyzed with an R script, which is available at GitHub (https://github.com/richard02050411/Calcium-Flux-FACs/blob/master/R-code).

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

## Acknowledgements

We thank J. Zikherman for her constructive suggestions regarding this paper. We thank M. Krangel and A. Naik for their valuable consultations of methodology for TCR repertoire analysis. We thank A. Roque for animal husbandry, J. Mueller and W. Lo for irradiation of mice, UCSF Parnassus Flow Cytometry Core for maintaining FACSArias instruments and services, the Gladstone transgenic core facility for generating *Nir3*[−/−] mice, W. Lo for testing gRNA in EL4 cells and NIH Tetramer Core Facility for providing the H-2Kb OVA tetramers. This work was supported by in part by the Howard Hughes Medical Institute, the NIH (grants NIAID R37 AI114575 (A.W.), NIGMS R01 GM 144479 (J.L.) and NIAMS K01 AR065481 (B.B.A.)) and DRC Center Grant P30 DK063720 (UCSF Parnassus Flow Cytometry Core, UCSF Diabetes Center).

## Author contributions

W.L., Y.A.H. and A.W. conceived the project and designed experiments. B.B.A. created the *Nir3*$^{-/-}$ mice. W.L. and Y.A.H. performed experiments and analyzed data. W.L. and K.S. designed and performed the phylogenetic study. B.B.A. and J.L. assisted with data interpretation and experimental design. W.L. and A.W. wrote the manuscript. A.W. supervised the study.

## Competing interests

The authors declare no competing interests.

## Additional information

**Extended data** is available for this paper at https://doi.org/10.1038/s41590-022-01372-2.

**Correspondence and requests for materials** should be addressed to Arthur Weiss.

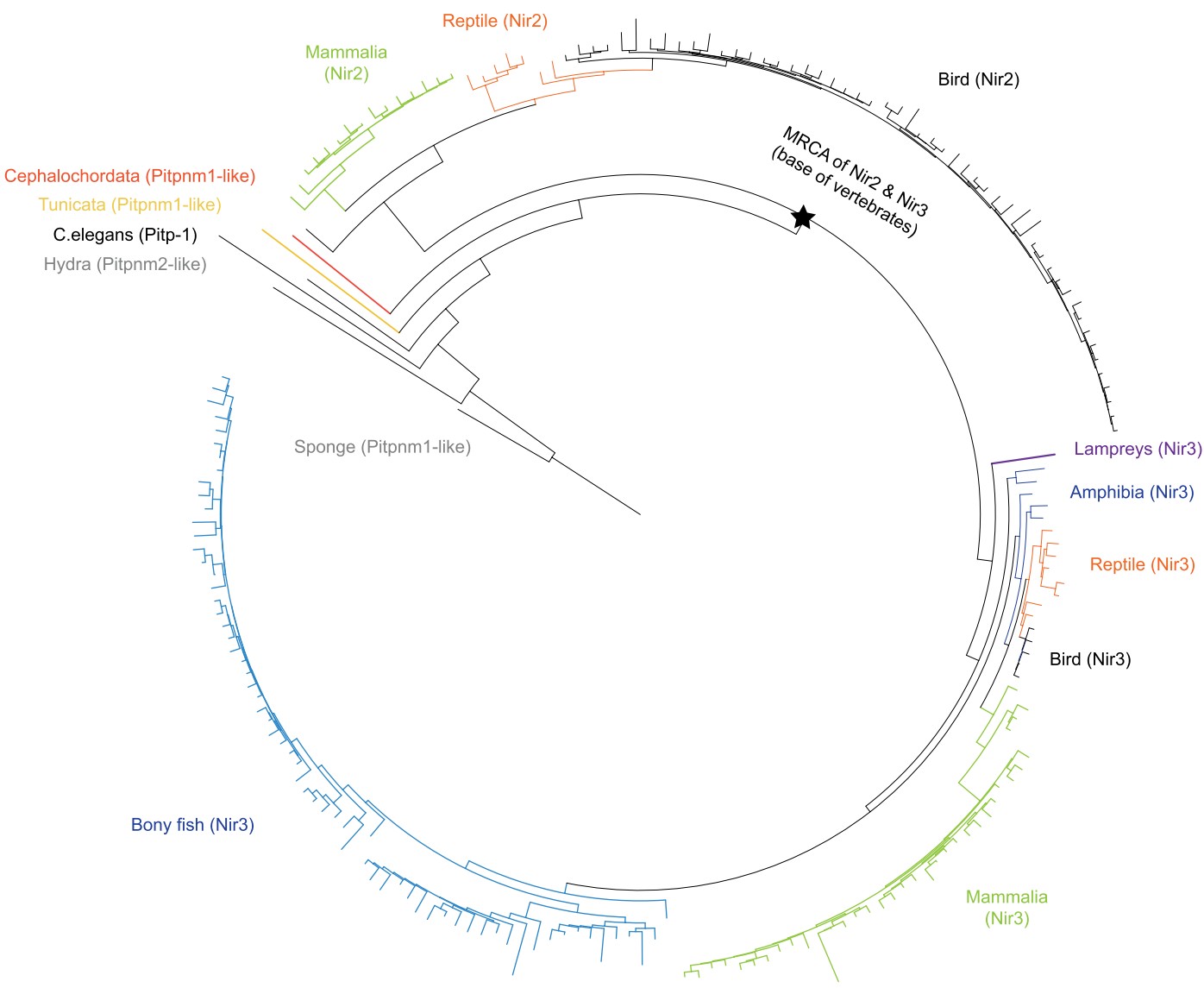

**Extended Data Fig. 1 | Evolution of Nir2 and Nir3.** The tree indicates the reconstructed phylogenetic relationships between the Nir2 and Nir3 proteins with an emphasis on vertebrate lineages. The most recent common ancestor of these orthologs is represented with a star at the base of vertebrates. See Methods for details on sequence analysis.

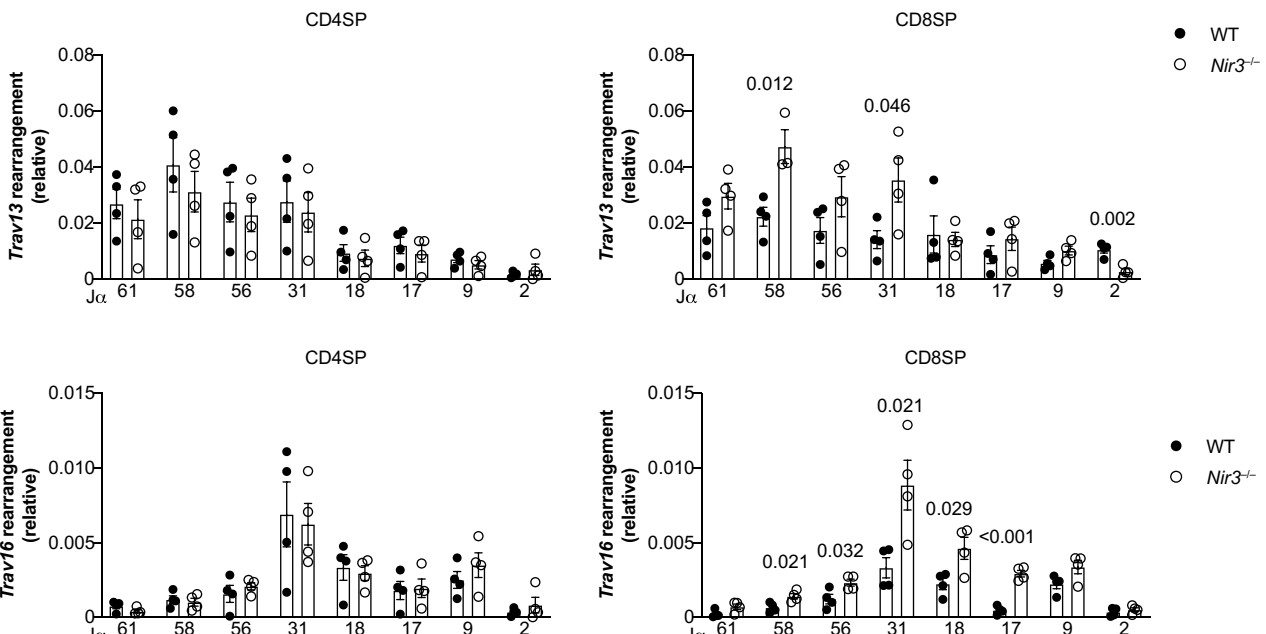

**Extended Data Fig. 2 | Altered Vα-Jα rearrangements in *Nir3*[−/−] mice.**
Rearrangement of selected Vα segments to different Jα segments in CD4SP and CD8SP thymocytes were assessed by SYBR Green-qPCR with normalization to Cd14. N = 4 mice. Data are shown as the mean ± s.e.m. and are representative of two independent experiments. *P* values were determined using a two-tailed Student's *t*-test.

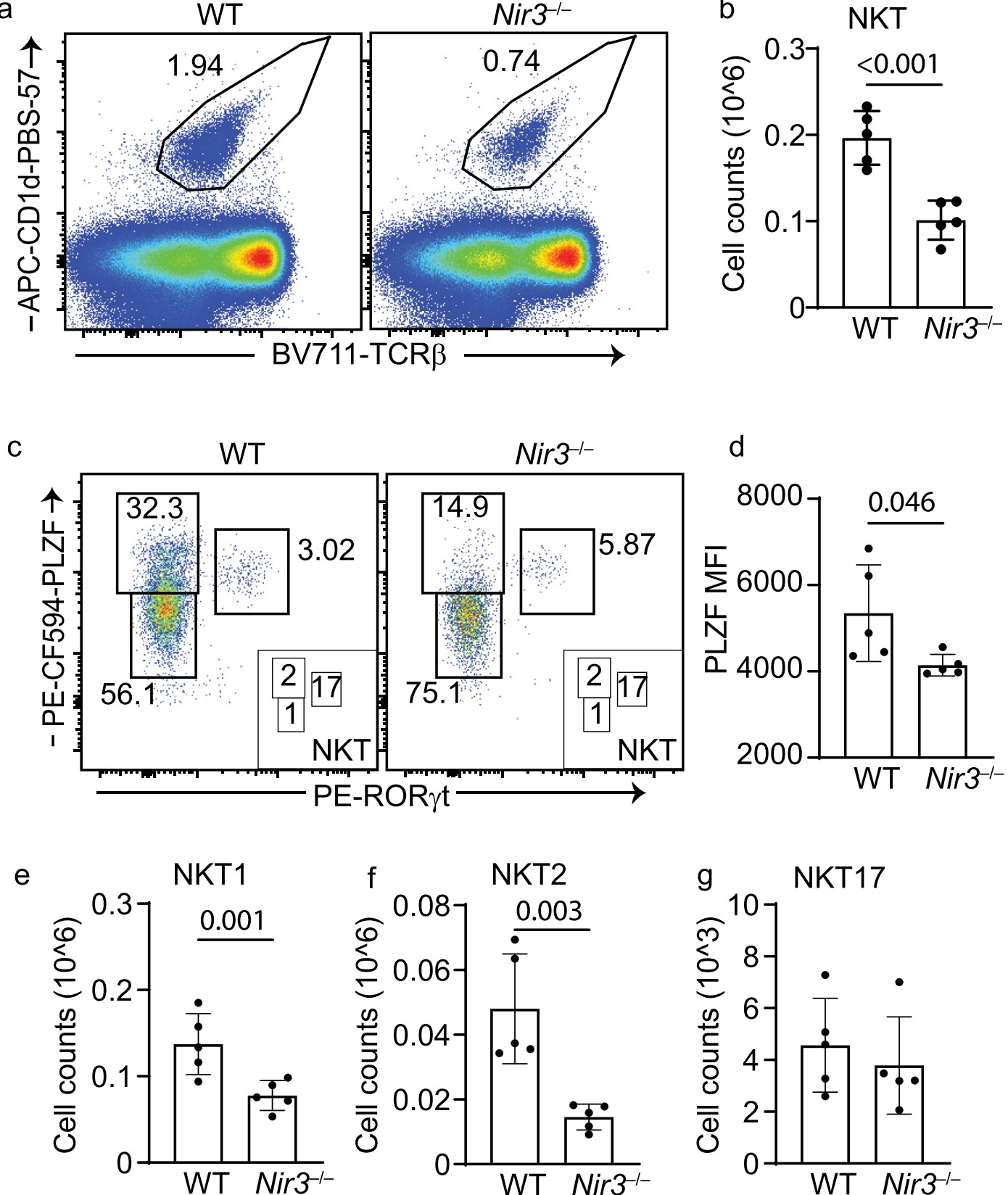

**Extended Data Fig. 3 | Decreased thymic NKT cells in *Nir3*⁻/⁻ mice.** (a) Representative staining of NKT cells gated as TCRβ^int, CD1d-PBS-57 tetramer+ cells in live, CD8-negative thymocytes from WT and *Nir3*⁻/⁻ thymus. (b) Absolute numbers of thymic NKT cells in the WT and *Nir3*⁻/⁻ mice. (c) NKT cell subsets were gated as PLZF^lo RORγt⁻ (NKT1), PLZF^hi RORγt⁻ (NKT2) and PLZF^int RORγt⁺ (NKT17). (d) PLZF expression levels in total WT and *Nir3*⁻/⁻ NKT cells. (e-g) Absolute numbers of the NKT cell subsets in WT and *Nir3*⁻/⁻ mice. N = 5 mice. Data are shown as the mean ± s.e.m. and are representative of two independent experiments. *P* values were determined using a two-tailed Student's *t*-test.

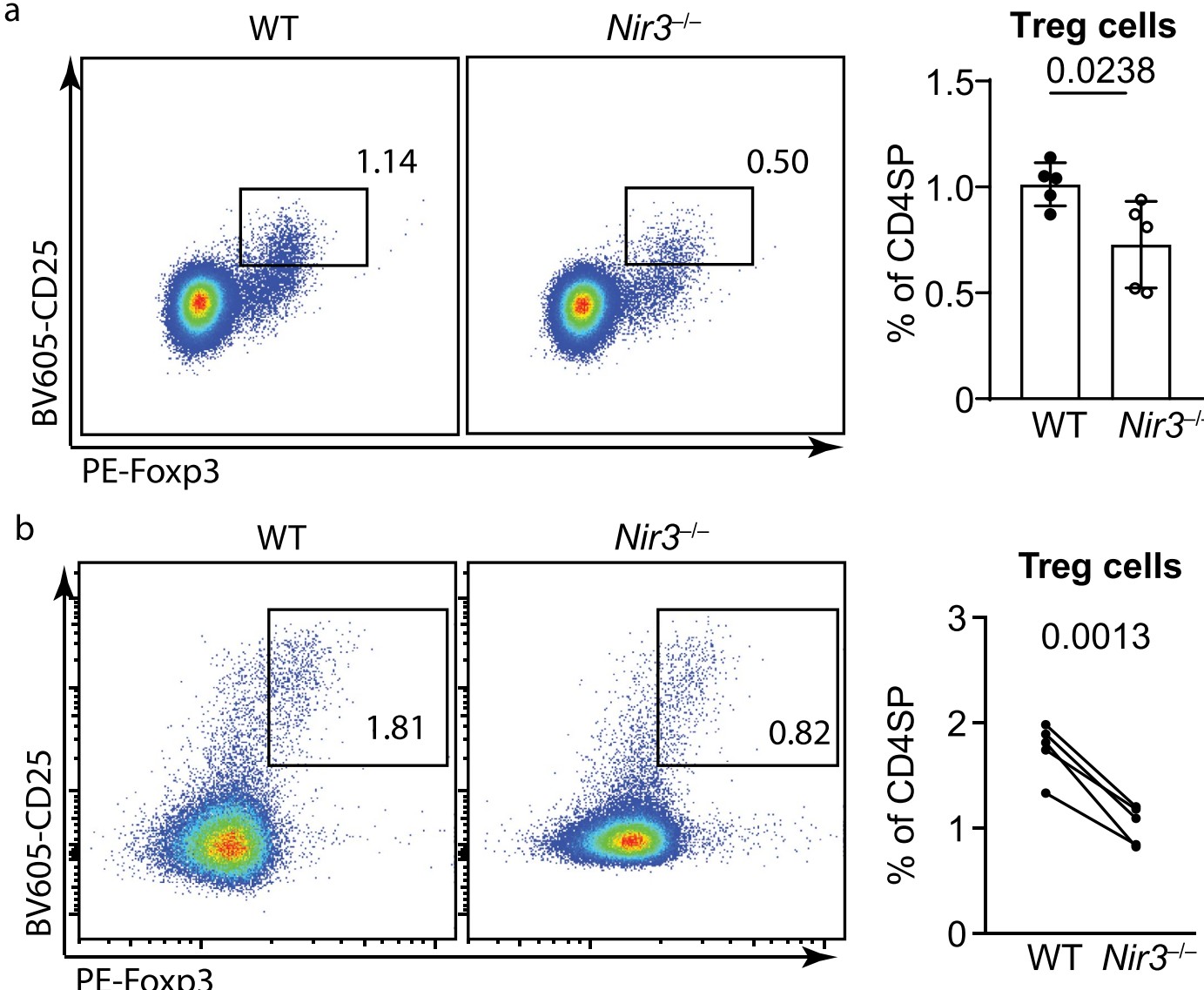

**Extended Data Fig. 4 | Decreased thymic Treg cells in *Nir3*[−/−] mice.**
CD25 + Foxp3+ Treg cells were gated from CD4SP thymocytes in WT and *Nir3*[−/−] mice (a) or bone marrow chimera mice (b). Representative flow plots (left) and quantification (right) of percentages of Treg cells were shown. WT and *Nir3*[−/−] donor cells recovered from the same recipient mouse were connected with lines (b). N = 5 mice. Data are shown as the mean ± s.e.m. and are representative of at least 2 independent experiments. *P* values were determined using a two-tailed Student's *t*-test.

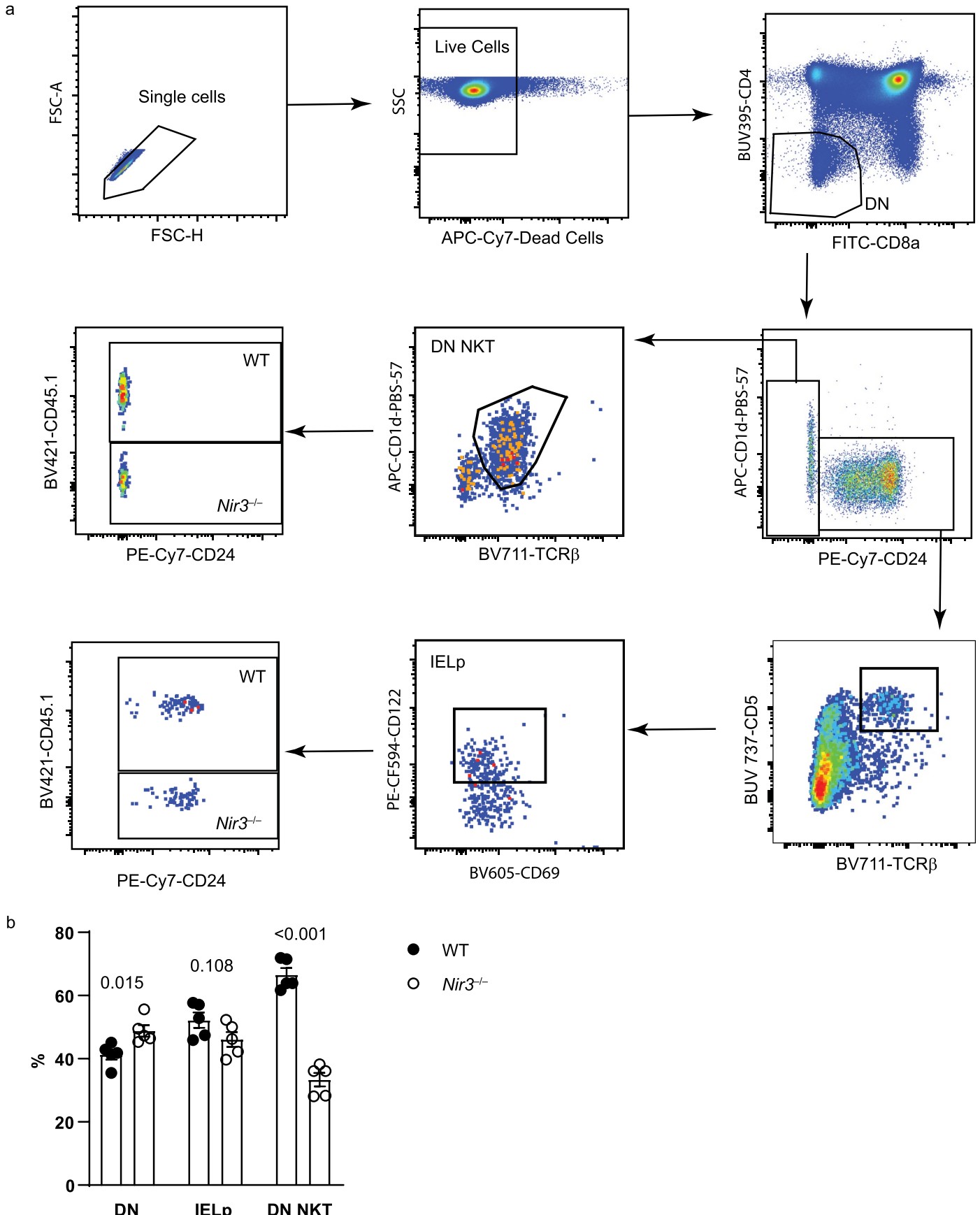

**Extended Data Fig. 5 | Nir3 is dispensable for IELp development.** Bone marrow cells from CD45.1/CD45.2 Nir3 sufficient mice (WT) and CD45.2 *Nir3*[−/−] mice (*Nir3*[−/−]) were mixed at a 1:1 ratio and adoptively transferred into CD45.1 hosts for six weeks. (a) DN NKT cells and IEL precursors (IELp) are gated from DN thymocytes. (b) The percentages of WT and *Nir3*[−/−] cells in each subset were quantified. N = 5 mice. Data are shown as the mean ± s.e.m. and are representative of 2 independent experiments. *P* values were determined using a two-tailed Student's *t*-test.

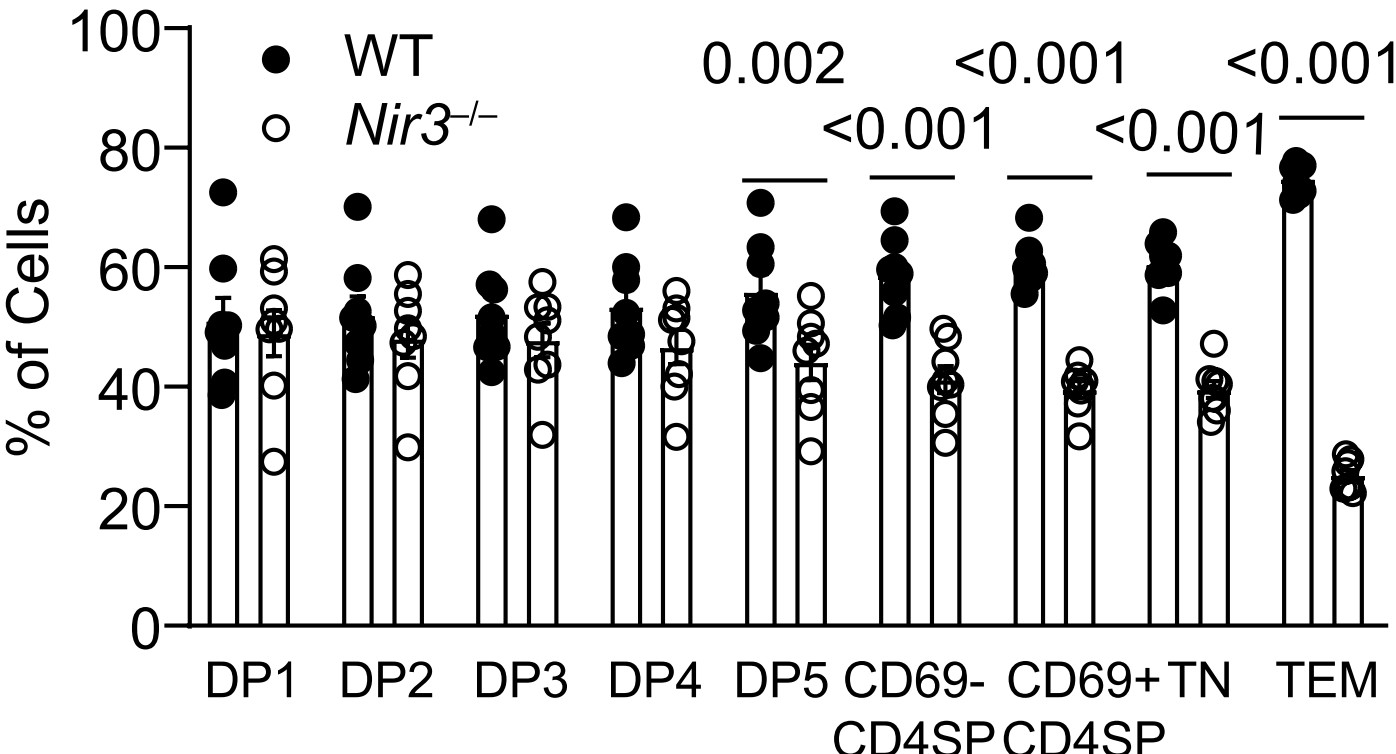

**Extended Data Fig. 6 | Impaired development of *Nir3*⁻/⁻ OT-II transgenic T cells.** Bone marrow cells from CD45.1/CD45.2 Nir3 sufficient (WT) OT-II mice and CD45.2 *Nir3*⁻/⁻ OT-II mice were mixed at 1:1 ratio and adoptively transferred into CD45.1 hosts. 6 weeks after the transfer, WT OT-II and *Nir3*⁻/⁻ OT-II thymocytes at each development stage and splenic T cells were identified by congenic CD45 markers. The percentages of WT OT-II cells and *Nir3*⁻/⁻ OT-II cells in each stage were quantified. TN, T naive; TEM, T effector memory-like. N = 8 mice. Data are shown as the mean ± s.e.m. and are pooled from 2 independent experiments. *P* values were determined using a two-tailed Student's *t*-test.

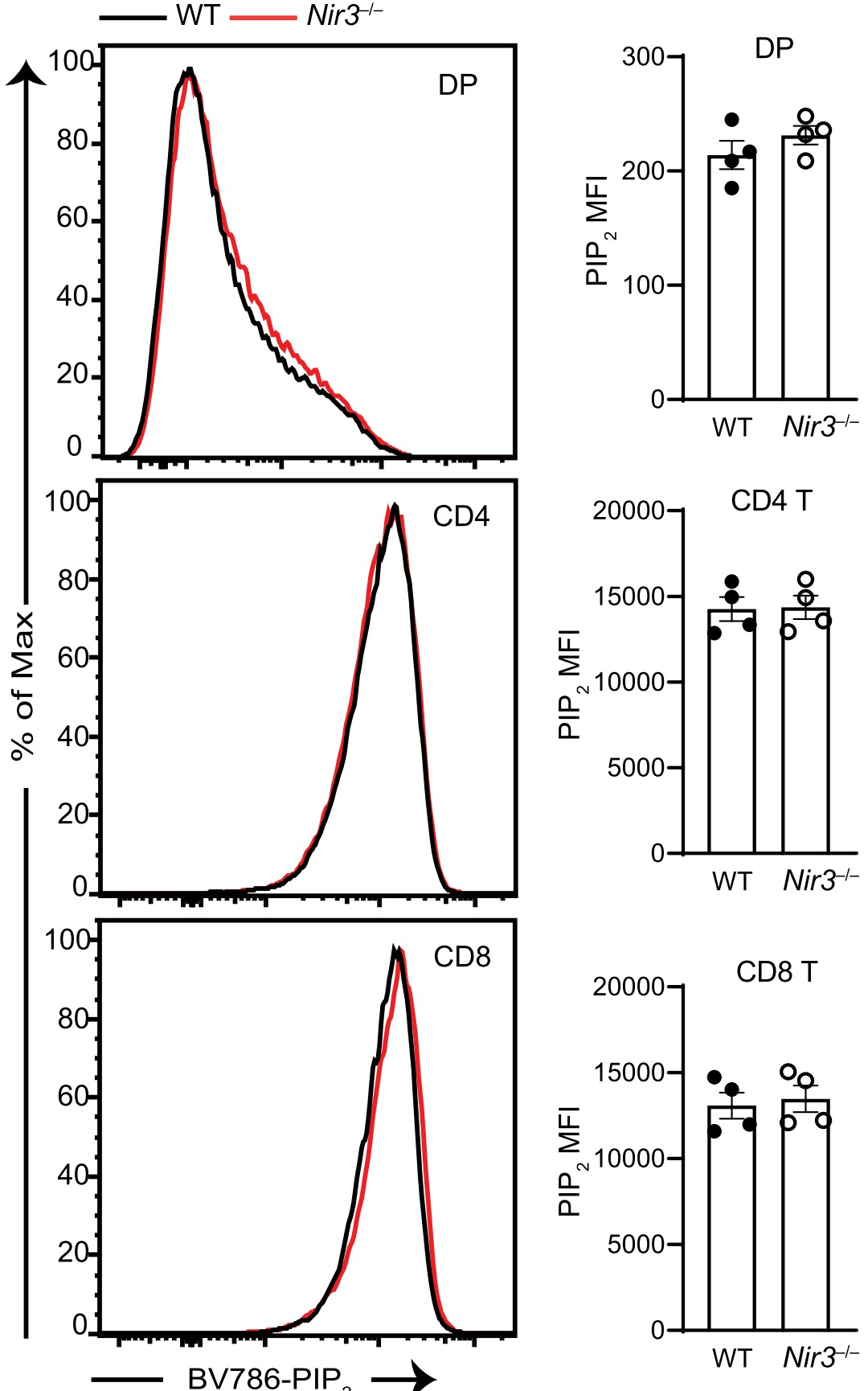

**Extended Data Fig. 7 | Nir3 is dispensable for maintaining PIP$_2$ levels in quiescent T cells.** PIP$_2$ levels on DP thymocytes (DP), CD4 (CD4 T) and CD8 (CD8 T) splenic T cells were measured by flow cytometry. N = 4 mice. Data are shown as the mean ± s.e.m. and are representative of two independent experiments.

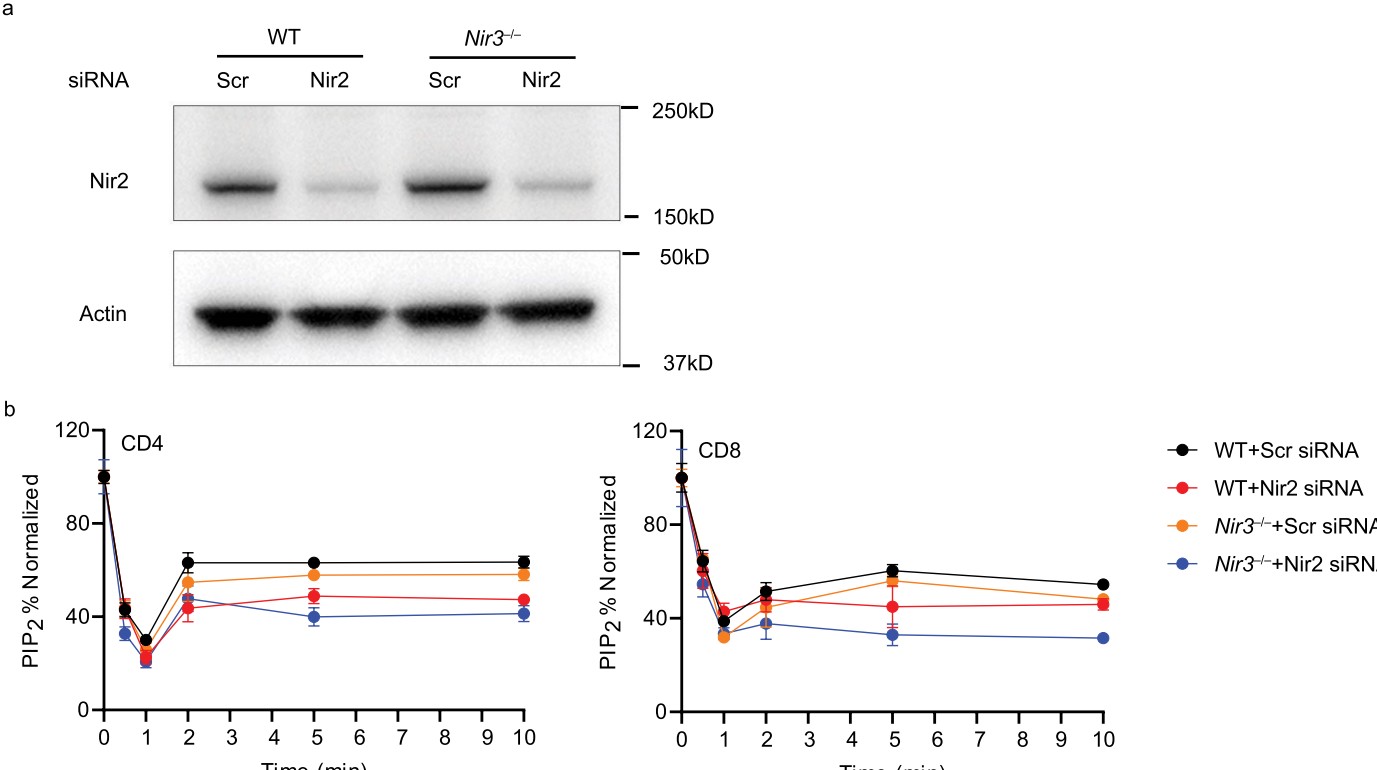

**Extended Data Fig. 8 | Nir2 regulates PIP$_2$ replenishment in mature T cells.** Wildtype (WT) and *Nir3*$^{-/-}$ CD3 + T cells were stimulated with bead-bound anti-CD3 and anti-CD28 antibodies for 3 days, transfected with scramble (Scr) or Nir2 siRNAs and rested in IL-7 containing media for 2 days. (a) Nir2 protein levels were assessed by immunoblots 2 days post transfection. (b) Cells were stimulated with 10μg/ml anti-CD3 antibodies (clone 2c11). PIP$_2$ abundances at indicated time points were assessed by flow cytometry. The data are shown as the mean ± s.e.m. N = 3. Data are representative of two independent experiments.

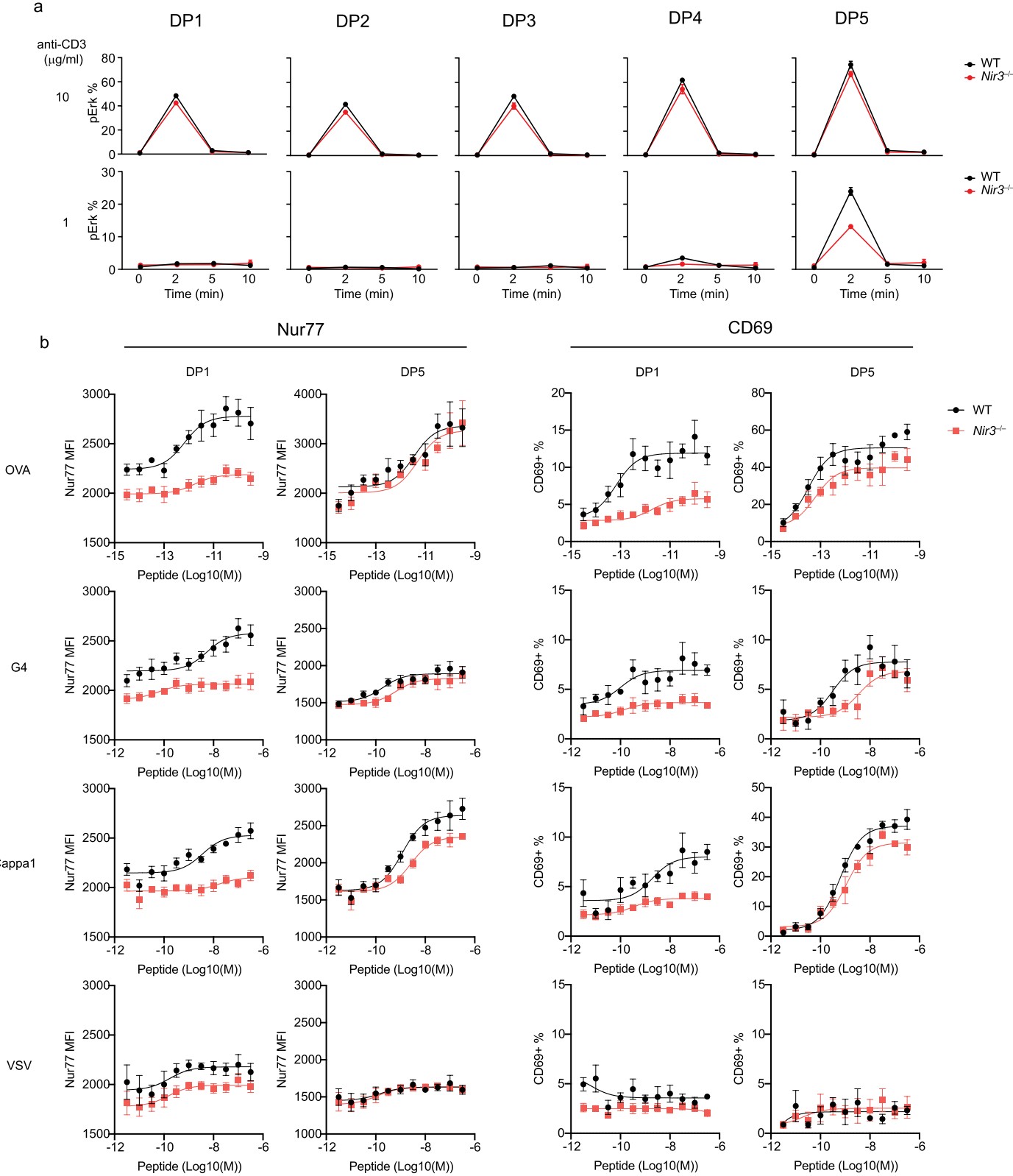

**Extended Data Fig. 9 | Nir3 modulates Erk and Nur77 activation induced by weak TCR stimulation.** (a) WT or $Nir3^{-/-}$ DP thymocytes were stimulated with indicated amount of anti-CD3 at 0 min. Erk phosphorylation were assessed by flow cytometry. N = 3. The data are shown as the mean ± s.e.m. (b) WT or $Nir3^{-/-}$ OT-I DP1 and DP5 thymocytes were sorted by flow cytometry and co-cultured with full agonist (OVA), weak agonist (G4), positive selecting (Cappa1) and non-binding (VSV) peptide variants. After 3 hours, Nur77 expression and the percentage of thymocytes expressing CD69 were measured. N = 4. The data are shown as the mean ± s.e.m. with fitted curve (three parameters). Data are representative of two independent experiments.

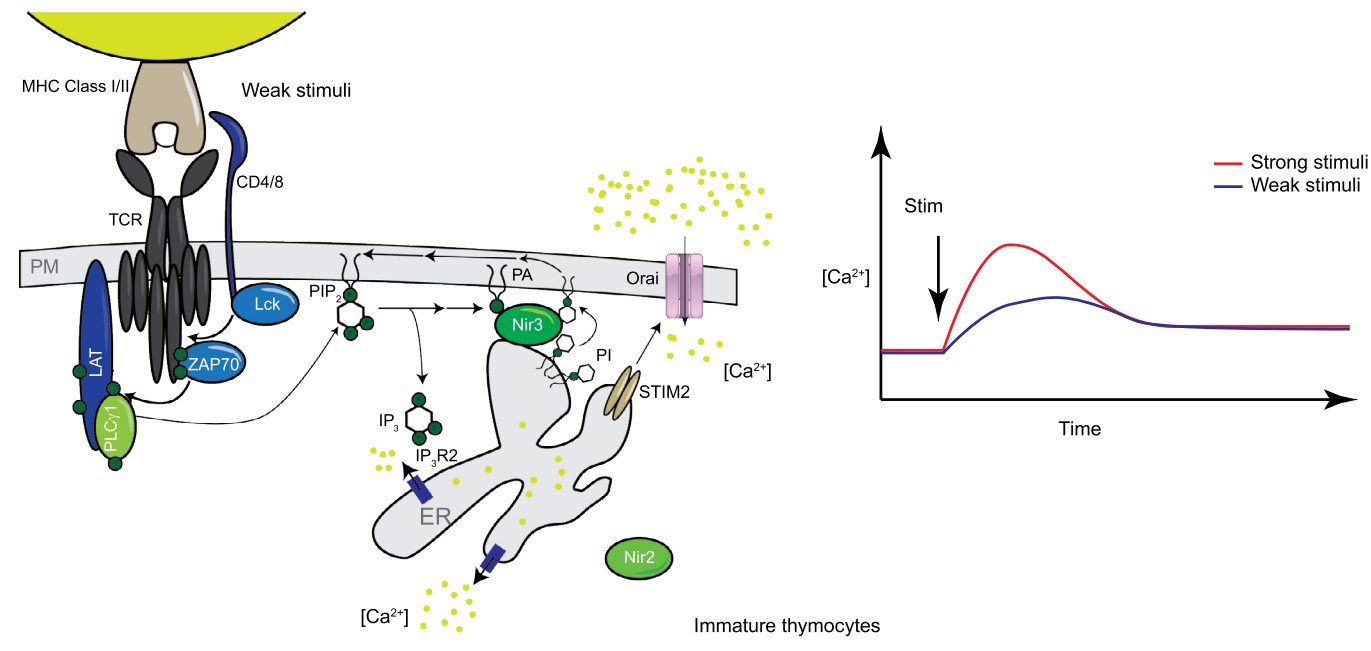

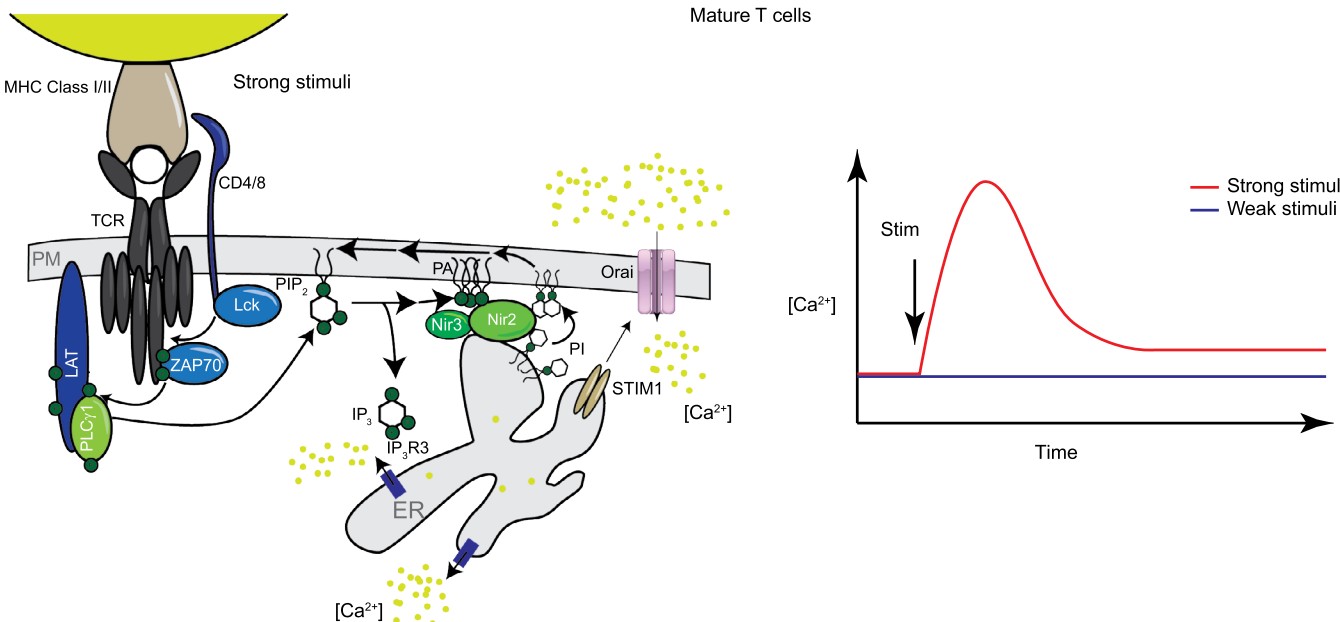

**Extended Data Fig. 10 | Proposed Model.** In immature thymocytes, Nir3, IP$_3$R2 and STIM2 are predominantly expressed. Upon weak TCR stimulation from self pMHC complex on thymic stromal cells, a small fraction of PIP$_2$ is cleaved by PLCγ1 which generates limited amounts of PA at the ER-PM junctions. Nir3 is recruited to the junctions and transfers PI to the PM which is important for PIP$_2$ replenishment. The whole Nir3-IP$_3$R2-STIM2 circuit is critical for the sustained TCR signaling upon weak TCR stimuli. In mature T cells. Nir2, IP$_3$R3 and STIM1 are major components of the SOCE pathway. The low sensitivity of the Nir2-IP$_3$R3-STIM1 contributes to the non-responsiveness of mature T cells to weak TCR stimuli. Upon strong TCR stimulation from foreign peptide-MHC ligands, a large amount of PIP$_2$ is metabolized which produces increased PA at the ER-PM junctions. Both Nir2 and Nir3 are recruited to the junctions in tandem. Nir2 transfers PI more efficiently than Nir3. Nir2 is the main PI transfer protein and Nir3 is less required for the PIP$_2$ replenishment in this context.

# Reporting Summary

## Statistics

For all statistical analyses, confirm that the following items are present in the figure legend, table legend, main text, or Methods section.

| n/a | Confirmed | |
|---|---|---|
| ☐ | ☒ | The exact sample size (*n*) for each experimental group/condition, given as a discrete number and unit of measurement |
| ☐ | ☒ | A statement on whether measurements were taken from distinct samples or whether the same sample was measured repeatedly |
| ☐ | ☒ | The statistical test(s) used AND whether they are one- or two-sided<br>*Only common tests should be described solely by name; describe more complex techniques in the Methods section.* |
| ☒ | ☐ | A description of all covariates tested |
| ☐ | ☒ | A description of any assumptions or corrections, such as tests of normality and adjustment for multiple comparisons |
| ☐ | ☒ | A full description of the statistical parameters including central tendency (e.g. means) or other basic estimates (e.g. regression coefficient) AND variation (e.g. standard deviation) or associated estimates of uncertainty (e.g. confidence intervals) |
| ☐ | ☒ | For null hypothesis testing, the test statistic (e.g. *F*, *t*, *r*) with confidence intervals, effect sizes, degrees of freedom and *P* value noted<br>*Give P values as exact values whenever suitable.* |
| ☒ | ☐ | For Bayesian analysis, information on the choice of priors and Markov chain Monte Carlo settings |
| ☒ | ☐ | For hierarchical and complex designs, identification of the appropriate level for tests and full reporting of outcomes |
| ☒ | ☐ | Estimates of effect sizes (e.g. Cohen's *d*, Pearson's *r*), indicating how they were calculated |

*Our web collection on statistics for biologists contains articles on many of the points above.*

## Software and code

Policy information about availability of computer code

| | |
|---|---|
| Data collection | The flow cytometry data were collected via BD FACSDiva v8.0.1 software. The TIRF images were collected via Nikon NIS-Elements AR imaging software v5.41.02 with A1 and C2 confocal applications |
| Data analysis | The flow cytometry data were analyzed by BD FLOWJO software version 10.4. In calcium flux experiments, FACS data were exported from FlowJo in CSV format and analyzed with an R script (https://github.com/richard02050411/Calcium-Flux-FACs/blob/master/R-code) by R version 4.0. Graphs and statistics were generated and analyzed using Graphpad Prism software v8.4. The TIRF images were analysis by ImageJ2 v2.3.0/1.53p.<br>In phylogenetic analysis of Nir2 and Nir3, MAFFT was employed to construct a multiple sequence alignment (MSA) for the list of sequences. TrimAI was used to trim for regions with more than 20% gaps and the trimmed MSA was used as input for phylogenetic reconstruction by maximum likelihood analyses through FastTree MP. The initial alignment covered over 750 sequences across 600 species and Phylogenetic Diversity Analyzer was used to generate a sub-sampled sequence list that simultaneously preserved tree features but with an emphasis on the vertebrate lineage with select but well-spaced outgroups including chordates, invertebrates, and sponges (totally 200 sequences). The sub-sampled list was subsequently aligned, trimmed, and a tree was generated as described above, and annotated manually using the ITOL website. |

For manuscripts utilizing custom algorithms or software that are central to the research but not yet described in published literature, software must be made available to editors and reviewers. We strongly encourage code deposition in a community repository (e.g. GitHub). See the Nature Portfolio guidelines for submitting code & software for further information.

## Data

Policy information about availability of data

All manuscripts must include a data availability statement. This statement should provide the following information, where applicable:

- Accession codes, unique identifiers, or web links for publicly available datasets
- A description of any restrictions on data availability
- For clinical datasets or third party data, please ensure that the statement adheres to our policy

RNA expression profiles of Nir2 and Nir3 in human tissues and mouse T cell populations are available on The Human Protein Atlas (https://www.proteinatlas.org/) and Immunological Genome Project (https://www.immgen.org/), respectively.

# Field-specific reporting

Please select the one below that is the best fit for your research. If you are not sure, read the appropriate sections before making your selection.

☒ Life sciences          ☐ Behavioural & social sciences          ☐ Ecological, evolutionary & environmental sciences

For a reference copy of the document with all sections, see nature.com/documents/nr-reporting-summary-flat.pdf

# Life sciences study design

All studies must disclose on these points even when the disclosure is negative.

| | |
|---|---|
| Sample size | For primary mouse cells experiments, cells from the whole organ were used or cell numbers were as indicated in each method section. Cell numbers were decided upon empiricals, and were routinely used in the lab that were known to be sufficient to allow reliable detection of flow cytometry.  The animal numbers used in each experiment were indicated in the bar graphs or figure legends. No statistical methods were used to pre-determine sample sizes but our sample sizes are similar to those reported in previous publications (see methods for references). |
| Data exclusions | We did not exclude any sample. |
| Replication | The findings were reliably reproducible.  Each set of experiments was repeated at least for two times as indicated in the figure legends. Different experiments were designed to test the hypothesis from orthogonal perspectives.  For T cell developmental experiments, we utilized two different approaches (CRISPR knockout animals and bone marrow chimera animals) and obtained similar results. |
| Randomization | We did not use randomization in our experiments.  In animal experiments, age matched animals were allocated based on their genotypes.  In cell stimulation experiments, cells with the same genotype were pulled together and equally allocated into groups before receiving different stimulations. |
| Blinding | Data collection and analysis were not performed blind to the conditions of the experiment.  For T cell development experiments, group allocation was not applicable because mice were grouped based on and compared across different genotypes. |

# Reporting for specific materials, systems and methods

We require information from authors about some types of materials, experimental systems and methods used in many studies. Here, indicate whether each material, system or method listed is relevant to your study. If you are not sure if a list item applies to your research, read the appropriate section before selecting a response.

### Materials & experimental systems

| n/a | Involved in the study |
|---|---|
| ☐ | ☒ Antibodies |
| ☐ | ☒ Eukaryotic cell lines |
| ☒ | ☐ Palaeontology and archaeology |
| ☐ | ☒ Animals and other organisms |
| ☒ | ☐ Human research participants |
| ☒ | ☐ Clinical data |
| ☒ | ☐ Dual use research of concern |

### Methods

| n/a | Involved in the study |
|---|---|
| ☒ | ☐ ChIP-seq |
| ☐ | ☒ Flow cytometry |
| ☒ | ☐ MRI-based neuroimaging |

## Antibodies

| | |
|---|---|
| Antibodies used | Antibody name (clone, source, catalogue number, working dilution)<br>Nir2 Polyclonal antibody (Rabbit polyclonal, Proteintech, 26983-1-AP, 1:1000)<br>PITPNM2 / NIR3 Antibody (aa162-435) (Rabbit polyclonal, Lifespan Bio, LS-C178853, 1:1000)<br>Biotinylated Anti-PtdIns(4,5)P2 IgM (2C11, Echelon-inc, z-b045, 1:100) |

AffiniPure Goat Anti-Armenian Hamster IgG (H+L)  (Jackson ImmunoResearch Lab, 127-005-099, 1:100)
BD Pharmingen™ PE Rat anti-Mouse Foxp3 (MF23, BD Biosciences, 560408, 1:100)
Armenian Hamster anti-mouse CD3ε Antibody (2c11, Weiss lab,  1:100)
BD Pharmingen™ APC Rat Anti-Mouse CD5 (53-7.3, BD Biosciences, 550035, 1:100)
BD Horizon™ BUV395 Rat Anti-Mouse CD4 (RM4-4, BD Biosciences, 563790, 1:100)
BD Pharmingen™ FITC Rat Anti-Mouse CD8a (53 6.7, BD Biosciences, 553030, 1:100)
BD Horizon™ BUV737 Rat Anti-Mouse CD8a (53 6.7, BD Biosciences, 612759, 1:100)
APC-CD1d-PBS-57 tetramer  (NIH tetramer core , 1:100)
BD Horizon™ PE-CF594 Mouse Anti-PLZF (R17-809, BD Biosciences, 565738, 1:100)
BD Pharmingen™ PE Mouse anti-Mouse RORγt (Q31-378, BD Biosciences, 562607, 1:100)
BV605-Armenian monoclonal antimouse CD69 (H1.2F3, BD Biosciences, 563290, 1:100)
PE-Cy7-rat monoclonal antimouse CD24 (M1/69, BD Biosciences, 560536, 1:100)
rabbit monoclonal antimouse/human phopspho-p44/42 MAPK (Erk1/2) (Thr202/Tyr204) (197G2, Cell Signaling Technology, 4376, 1:100)
BD Horizon™ BV421 Mouse Anti-Mouse CD45.1 (A20, BD Biosciences, 563983, 1:100)
BD Pharmingen™ PE-Cy™7 Mouse Anti-Mouse CD45.2 (104, BD Biosciences, 560696, 1:100)
BV711-anti-mouse TCR β chain Antibody (H57-597, BD Biosciences, 563135, 1:100)
BV421-anti-mouse TCR β chain Antibody (H57-597, BD Biosciences, 562839, 1:100)
BD Horizon™ BUV737 Rat Anti-Mouse CD5 (53-7.3, BD Biosciences, 612809, 1:100)
PE-anti-mouse/rat/human CD27 Antibody (LG.3A10, BD Biosciences, 558754, 1:100)
BV605-anti-mouse CD25 Antibody (PC61, BD Biosciences, 563061, 1:100)
PE-Cy™7 Rat Anti-Mouse CD44 (IM7, BD Biosciences, 560569, 1:100)
BV605-anti-mouse CD62L Antibody (MEL-14, BD Biosciences, 563252, 1:100)
BD Horizon™ BV786 Streptavidin  (BD Biosciences, 563858, 1:100)
BD Horizon™ PE-CF594 Mouse Anti-Ki-67 (B56, BD Biosciences, 567120, 1:100)
BD Horizon™ PE-CF594 Rat Anti-Mouse CD122 (TM-β1, BD Biosciences, 564763, 1:100)
Nur77 Monoclonal Antibody (12.14), PE, eBioscience™ (12.14, ThermoFisher Scientific, 12-5965-82, 1:100)

Validation | All the antibodies are from commercial sources and have been validated by the vendors.  Validation data are available on the manufacturer's website as well as summarized in supplementary table 2.

# Eukaryotic cell lines

Policy information about cell lines

Cell line source(s) | HEK293 cells were obtained form UCSF cell culture facility.

Authentication | HEK293 cells were not authenticated

Mycoplasma contamination | The HEK293 cells have been tested for mycoplasma in past years.

Commonly misidentified lines (See ICLAC register) | HEK293 cell line was not listed in ICLAC database.

# Animals and other organisms

Policy information about studies involving animals; ARRIVE guidelines recommended for reporting animal research

Laboratory animals | Nir3 deficient mice were generated as described in the method section and backcrossed to C57BL/6 mice for at least six generations. BoyJ (CD45.1) mice (B6.SJL-Ptprca Pepcb/BoyJ) and OT-1 mice (C57BL/6-Tg(TcraTcrb)1100Mjb/J) were originally obtained from The Jackson Laboratory .  CD45.1/CD45.2 mice were generated by crossing BoyJ and C57BL/6 mice.  All mice used in these studies were housed in a specific pathogen-free facility at UCSF according to the University Animal Care Committee and National Institutes of Health (NIH) guidelines.  The temperature of the mouse room is between 68-79°F with 30-70% humidity.  A 12 light/12 dark cycle is used.

Wild animals | The study did not involve wild animals.

Field-collected samples | The study did not involve samples collected from the field.

Ethics oversight | Animals were maintained in accordance with Institutional Animal Care and Use Committee protocols, University of California, or approved by San Francisco veterinary committees, and are in accordance with NIH guidelines.

Note that full information on the approval of the study protocol must also be provided in the manuscript.

# Flow Cytometry

## Plots

Confirm that:

☒ The axis labels state the marker and fluorochrome used (e.g. CD4-FITC).

☒ The axis scales are clearly visible. Include numbers along axes only for bottom left plot of group (a 'group' is an analysis of identical markers).

☒ All plots are contour plots with outliers or pseudocolor plots.

☒ A numerical value for number of cells or percentage (with statistics) is provided.

## Methodology

| | |
|---|---|
| Sample preparation | Thymus, spleens and lymph node single cell suspensions were prepared by gently tweezing samples in cold PBS buffer containing 0.5% BSA and 0.2% EDTA. |
| Instrument | BD LSR Fortessa was used to collect flow cytometry data. |
| Software | BD FACSDiva v8.0.1 software was used to collect samples.  FlowJo v10.4 was used to analyze flow cytometry data. |
| Cell population abundance | Sorted DP1 and DP5 thymocytes were about 90% purity. |
| Gating strategy | Doublets were excluded using forward light-scatter gating followed by gating on lymphocytes based on FSC-SSC.  Dead cells were excluded by live/dead stain.  Additional gating strategies were provided in the figures (Fig. 2a, 2d, 2e, 2h, 3a, 3c, 3e, 4a, 4b, 4e, 4h and 5f) |

☒ Tick this box to confirm that a figure exemplifying the gating strategy is provided in the Supplementary Information.

