## [Peer Review File · Nature Immunology]

Peer Review Information

Journal: Nature Immunology

Manuscript Title: The phosphatidylinositol (PI)-transfer protein Nir3 promotes PI(4,5)P₂ replenishment in response to TCR signaling during T cell development and survival

Corresponding author name(s): Arthur Weiss

Reviewer Comments & Decisions:

Decision Letter, initial version:

Subject: Decision on Nature Immunology submission NI-A33810

Message: 12th Apr 2022

Dear Art,

Your Article, "The phosphatidylinositol (PI)-transfer protein Nir3 promotes PI(4,5)P₂ replenishment in response to TCR signaling during thymocyte development and peripheral T cell survival" has now been seen by 3 referees. You will see from their comments copied below that while they find your work of considerable potential interest, they have raised quite substantial concerns that must be addressed. In light of these comments, we cannot accept the current manuscript for publication, but would be very interested in considering a revised version that addresses these serious concerns.

We can further discuss the revisions to the manuscript when you are ready to outline a response.

We hope you will find the referees' comments useful as you decide how to proceed. If you wish to submit a substantially revised manuscript, please bear in mind that we will be reluctant to approach the referees again in the absence of major revisions.

* If you have not done so already please begin to revise your manuscript so that it conforms to our Article format instructions at <http://www.nature.com/ni/authors/index.html>. Refer also to any guidelines provided in this letter.

The Reporting Summary can be found here:
<https://www.nature.com/documents/nr-reporting-summary.pdf>

You may use the link below to submit your revised manuscript and related files:
[REDACTED]

If you wish to submit a suitably revised manuscript we would hope to receive it within 6 months. If you cannot send it within this time, please let us know. We will be happy to consider your revision so long as nothing similar has been accepted for publication at Nature Immunology or published elsewhere.

Nature Immunology is committed to improving transparency in authorship. As part of our efforts in this direction, we are now requesting that all authors identified as 'corresponding author' on published papers create and link their Open Researcher and Contributor Identifier (ORCID) with their account on the Manuscript Tracking System (MTS), prior to acceptance. ORCID helps the scientific community achieve unambiguous attribution of all scholarly contributions. You can create and link your ORCID from the home page of the MTS by clicking on 'Modify my Springer Nature account'. For more information please visit www.springernature.com/orcid.

Thank you for the opportunity to review your work.

Kind regards,

Laurie

Laurie A. Dempsey, Ph.D.
Senior Editor
Nature Immunology
l.dempsey@us.nature.com
ORCID: 0000-0002-3304-796X

Referee expertise:

Referee #1: Proximal TCR signaling

Referee #2: TCR signaling

Referee #3: Calcium signaling

Reviewers' Comments:

Reviewer #1:

Remarks to the Author:

In this study the authors discovered in T cells the essential roles of Nir3, a member of phosphatidylinositol transfer proteins or PITPs which are implicated in traffic of phosphatidylinositol (PI) between intracellular membranes and the plasma membrane. TCR signaling in thymocytes and peripheral T cells requires production of messenger molecules IP3 and DAG with dependence on breakdown of PIP2. This study presented interesting findings that Nir3 significantly modulates replenishment of PIP2 and TCR-induced calcium signaling. The authors show that in the thymus the level of Nir3 elevates in correlation with β selection and positive selection, and the level of PIP2 is substantially elevated during thymocyte maturation. Loss of Nir3 results in obvious defects of thymocyte selections in TCR transgenic mice and normal non-transgenic mice. Such defects in T cell development are also found in competitive bone marrow transplant model. The authors also report that Nir3 is required for mature T cell survival in the periphery in cell-intrinsic manner. During T cell stimulation using anti-CD3, the replenish of PIP2 is significantly defective in Nir3 knockout T cells. In this study, the analyses are properly performed and the novel findings are important for understanding of T cell biology since the specific gene of the PITP family regulating PIP2 turnover during T cell development has not been known before. The manuscript is well prepared and conclusions are valid based on the current data. Identification of Nir3 as a target to regulate T cell response could lead to improved T cell therapies. Here are a few suggestions to improve the current manuscript for more complete view of the molecular basis of PIP2 replenishment in T cells through Nir3.

1. In Figure 1, the authors showed that the expression of Nir3 is restricted to thymus, and hardly detectable in the spleen in human samples while Nir2 expression look quite more universal. But in the mouse samples, Nir3 and Nir2 are quite similar in expression level in the T cells of spleen, regardless of the facts that Nir3 elevations associate with selection stages and Nir2 expression does not. Since Nir2 and Nir3 are highly conserved in amino acid sequences, it is difficult to understand that Nir2 is not taking a redundant role in peripheral T cells. In Figure 2 h, i and j, the authors showed that PIP2 elevates during thymocyte maturation but found Nir3 is dispensable for maintaining PIP2 levels in quiescent T cells which include splenic CD4 and CD8 T cells. Therefore, it is interesting to elucidate if in murine or human T cell line Nir3 and Nir2 are functionally redundant by performing knockout experiments using CRISPR/Cas9 system.
2. In many cases, TCR signaling defects would result in faulty thymic selection and T cell in periphery could display upregulation of CD44 and higher frequencies of memory T cells. In this study, the memory T cell frequencies are significantly lower as shown in Figure 5 g and h, which resembling phenotype of CD28 deficiency even though CD28 knockout mice do not have lymphopenia. I would suggest the authors to compare if Nir3 knockout impairs CD28 co-stimulation.
3. The authors revealed the critical role of Nir3 for T cell activation. It will be of interest for T cell biologists to analyze whether Nir3 is phosphorylated upon TCR stimulation. Nir3 contains 9 serine residuals that can be putatively phosphorylated according the public database.
4. The absolute numbers of cells instead of the % of cells need to be shown in Figures 3, 4 and 5. For instance, in panel h of Figure 4, the decrease in the % of DP5 might solely reflect the presence of higher numbers of DP1-4 cells.

Reviewer #2:

Remarks to the Author:

The manuscript by Lu, et al characterizes the regulation of PI(4,5)P2 (referred to as PIP2) in the plasma membrane of thymocytes and peripheral T cells. The authors describe a protein called Nir3 that is important in regenerating PIP2 when it is consumed by T cell receptor signaling leading to the activation of PLC- γ 1. The authors show that deletion of Nir3 leads to modest defects in T cell development. There is a partial block at the pre-TCR signaling stage and a small reduction in the proportion of CD4 SP thymocytes. Peripheral T cell numbers are also modestly reduced. When DP thymocytes were stimulated, Nir3-deficient cells had impaired calcium responses and modestly impaired regeneration of PIP2, particularly following weaker TCR stimulation. Based on these findings, the authors propose that Nir3 plays a critical role in PIP2 replenishment during TCR signaling.

This manuscript addresses a novel mechanism involved in TCR signaling leading to the activation of PLC- γ 1. They provide convincing evidence that, in the absence of Nir3, there are modest defects in PIP2 replenishment during TCR signaling, and that this, in turn, leads to modest impairments in T cell development and peripheral T cell homeostasis. However, the overall magnitude of the effects of Nir3-deficiency are quite small; furthermore, none of the data address any robust biological role for this pathway, such as impacting T cell responses to infection or to antigenic challenge.

In addition, there are specific issues the authors should address to strengthen the data presented.

1. The flow cytometry data in figures 3 and 4 require absolute numbers in addition to the percentages of subsets shown. This information is critical to interpreting any effects on T cell development.
2. The discussion of Nir gene phylogeny (Figure 1C) does not contribute significantly to the main message of the manuscript and is probably more appropriate as a supplemental figure. For instance, this analysis is not mentioned in the abstract.
3. The Ki67 staining is not convincing; consequently, it is difficult to conclude that WT and Nir3-deficient cells are different in regard to cell proliferation.
4. The authors suggest the importance of Nir3 in weak signaling conditions, but this is not consistently reflected or fully demonstrated in the figures. For instance, the weak vs. strong (N4 vs T4 vs G4) stimulation data from the supplement (Supp Figure 4b) do not make a compelling case for this conclusion. Additionally, the effect of Nir3 deficiency on NKT cell development also goes against this model, as evidence indicates that NKT cells undergo agonist selection and require strong TCR signal strength during development in the thymus. Therefore, these data are inconsistent with other data in the manuscript which support the claim that Nir3 is primarily important for selection of cells receiving weak and transient signaling.

Minor comments:

1. Figures are inconsistently labeled – some panels show 'ns' for comparisons not significantly different and other panels omit this.
2. It would be preferable to label the panels 'Nir3^{-/-}' rather than 'Nir3' for clarity.
3. The concluding sentence for the second Results section describing the data in Figure 2 is overstated, and should be toned down.
4. Better labeling or description of the images in Figure 6 would be helpful.
5. P-Erk data and stimulation with OVA peptide variants now in the supplement would be better as a main manuscript figure.

Reviewer #3:

Remarks to the Author:

This manuscript shows an important role for Nir3 proteins in PIP2 replenishment necessary during TCR stimulation. The authors generated Nir3 KO animals showing modest defects in thymocyte development and mature T cell survival in the periphery. While the functions of the Nir family of proteins, including Nir1/2/3 in the transfer of PI from the ER to the PM, are well studied in multiple cell types, there has not been an investigation of its physiological role in T cell development and homeostasis. Hence, there is a significant novelty in the study, and overall, I have high enthusiasm for this study. However, to support the working model of Nir3 that the authors proposed in Suppl. Fig. 6, the following experiments are necessary:

1. The authors need to check if the TCR repertoire is altered in Nir3-deficient mice. This data would justify later their experiments with OT-I/II TCR transgenics.
2. In Fig. 2, to strengthen their hypothesis that Nir2 plays an important role in mature T cells, while Nir3 is important in developing T cells, the authors need to compare the expression of Nir2 and Nir3 proteins between DP1-5 subpopulations, as well as mature T cells.

3. In Suppl. Fig. 3, the authors show reduced NKT1 and NKT2 cell populations in Nir3 KO mice. What is the effect of Nir3 deletion in the development of other agonist-selected T cell populations, including Tregs and intestinal intraepithelial lymphocytes? This result will consolidate the authors' hypothesis further.
 4. In Fig. 4, the authors need to add phenotypes of bone marrow chimeras with OT-II TCR transgenics, especially since the global KO of Nir3 showed phenotype in CD4+ SP T cells and not CD8 (Fig. 3b) in the thymus.
 5. In Fig. 5l-m, Nir3 KO cells show a defect in cell survival upon transfer, have the authors investigated which survival pathways (e.g., anti-apoptotic) are influenced in the KO?
 6. In relation to Fig. 5, have the authors checked the homeostatic proliferation of Nir3 KO naïve T cells in lymphopenic hosts?
 7. Except for mRNA analyses across different T cell subpopulations, the authors do not show any evidence of the specific function of Nir2 in mature T cells v/s Nir3 in developing T cells. Have the authors checked if the deletion of Nir2 in mature T cells (using sgRNAs) has a more substantial effect than that of Nir3 deletion? The authors need to compare phenotypes of deletion of Nir2, Nir3, and both in mature T cells in terms of their responses to Ca²⁺, PIP₂, and pERK levels. Does deletion of Nir2/3/both affect T cell polarization?
 8. Another critical aspect is regarding the functional redundancy between Nir2 and Nir3 in T cell functions. This can be achieved by reconstitution of Nir2 or Nir3 expression in Nir3 KO or Nir2/3 DKO T cells.
 9. In Figure 6, to further prove that the defects in PIP₂ replenishment cause the decrease in Ca²⁺ signaling, Ca²⁺ measurement using stimulations with thapsigargin or ionomycin (to bypass PLC β activity) needs to be shown.
 10. Any known links (or not) of Nir3 dysfunction to human diseases should be discussed.
- Minor point:
1. Add MW marker to Immunoblot in Suppl. Fig. 1C.
 2. The authors need to add details about the analysis of TIRF data in methods. Are intensities normalized to whole cells, or were individual ROIs within cells selected for normalization across multiple cells?

Author Rebuttal to Initial comments

Reviewer #1 (Remarks to the Author):

There are obvious improvements of the manuscript by Lu et al after revisions. For instance, the impact of Nir3 deficiency on CD28 co-stimulation was properly assessed. Even though the Nir2 and Nir3 knockout mutants were not obtained by CRISPR/Cas9 in T cell lines, the Nir2 knockdown by siRNA in OT-I cells (stimulated with OVA and variants) shows very clearly that Nir2 is essential for T cell activation. Moreover, Nir2 knockdown causes markedly reduced Nur77 expression accompanied by reduced PI(4,5)P₂ levels. The Nir2 knockdown cells had a more dramatic mutant phenotype than Nir3 knockout cells, suggesting indispensable roles of Nir2 in T cell activation. The authors answered the question concerning TCR induced phosphorylation of Nir3 by citing a published paper using the Jurkat cells expressing chimeric antigen receptor against CD19. The other experimental data added during the revision greatly improved the quality of the paper. The mutant phenotype of Nir3 knockout mice is not dramatic and primarily restricted to thymocyte development as noted by the other Reviewers. Despite this rather not spectacular phenotype, I still recommend publication of this paper because it is the first description of the role of this molecule in T cell development and for the in-depth functional analyses of

animal and cellular models that the manuscript contains. Here are a few more suggestions for the authors.

1. An analysis and discussion of the TCR induced phosphorylation of Nir3 (preferably with Nir2 analyzed in parallel) will constitute an important addition to provide more mechanistic insights into the unique role of Nir3. The current data support that Nir3 plays a critical role in PI(4,5)P2 replenishment following TCR engagement in thymocytes, and its function in T cell development is probably explained the elevated expression of Nir3 during thymocyte selections during which Nir2 is not abundantly expressed. However, in spleen (Figure 1b and 1c), both Nir2 and Nir3 are present and don't really display a very dramatic difference in protein abundance. In mature T cells loss of Nir3 leads to a very obvious defect in cell survival (Figure 6a). Accordingly, phenotype observed in Nir2 knockdown cells and Nir3 knockout cells is not convincingly only explained by the abundance of Nir2 and Nir3 proteins (Reviewer Figure 2c). Their activities to replenish PIP2 might be differentially regulated as well when T cells are stimulated.

We agree with the reviewer that the function of Nir2 and Nir3 may be regulated through posttranslational modifications including phosphorylation. In the previous communication, we have discussed that human Nir3 has 9 serine sites that can be putatively phosphorylated. Moreover, UniProt database shows 2 threonine and 15 serine residues in human Nir2 which can be putatively phosphorylated. In particular, Thr287 and Ser382 sites have been shown as substrates of CDK1, which could be important for T cell proliferation.

While it might be interesting to study these phosphorylation sites in Nir2 and Nir3 in T cells in the resting and stimulated states, we believe such an analysis would be technically difficult and addressing the function of such putative phosphorylation sites would be beyond the scope of this work. The analysis would be difficult due to lack of specific antibodies to Nir2 or Nir3 phosphorylation sites. Isolation of Nir2 or Nir3 in resting or stimulated primary thymocytes or T cells would be difficult. In particular, Nir3 is highly expressed in thymocytes but not in mature T cells. Due to the lower protein and ER content of primary thymocytes, detection of weak phosphorylation on those serine and threonine sites are beyond the capability of our lab and perhaps any laboratory. Doing such an analysis quantitatively seem extraordinarily difficult. Separating Nir2 or Nir3 localized in ER-PM junctions (perhaps the population of greatest interest) from the ER resident Nir2 or Nir3 protein subset would be challenging, to say the least. We suppose that if such incredibly challenging studies were feasible, the relevance of such resting or inducible changes in phosphorylation sites would have to be analyzed by mutagenesis. While these studies might add greatly to our understanding of Nir2 and Nir3 regulation and function, they really seem beyond the scope of this current study and could not be done in a timely manner.

2. As the authors noted, in lymphoreplete recipient mice, most of the adoptively transferred T cells had not undergone cell division, as reflected by their largely undiluted CellTrace dye (Figure 6d-e). Can this part be replaced by results from the same mice treated with OVA to visualize in vivo expansion of Nir3 knockout cells and their controls?

As we shown in the OVA peptide stimulation assays, Nir3 has been demonstrated important in self-ligand stimulation (tonic signaling), but not involved in agonist stimulation. Consistent with this hypothesis, Nir3^{-/-} T cells have a survival defect when transferred into naïve hosts (Figure 6a-c). We believe that this is likely mediated as a result of tonic TCR signaling in secondary lymphoid organs. Thus, we do not believe in vivo expansion induced by the agonistic OVA treatment is a better substitute for Figure 6d-f. We agree that most of transferred T cells had not undergone active cell proliferation in lymphoreplete recipient mice. We have provided a parallel experiment in lymphopenic recipient mice, where transferred T cells were actively proliferating, yet Nir3^{-/-} T cells proliferated less (see Figure 6g-i).

3. The descriptions and comparisons about Nir2 and Nir3 on the evolutionary basis is probably too lengthy. In the current manuscript, line 103 to 113, 513 to 540, 563 to 568 provides lots of information based on the phylogenetic analysis/protein domains. However, the functional analyses of Nir3 in the other species without Nir2 is not available, and the impact of Nir2 deficiency in mice for thymocyte development is not included in this study. A more concise introduction and discussion concerning such comparisons sound more appropriate.

We agree with reviewer #1 that the evolutionary comparisons in the Introduction and Discussion sections are unnecessarily long. Based on the latest comments from Reviewer #1 and previous comments from Reviewer #2, we realized that we put too much emphasis on the phylogenetic analysis of Nir2 and Nir3 in our original manuscript, which was not the focus of our experimental work. In order to make the manuscript more concise, we have shortened the Results section between lines 153 and 156 (highlighted in yellow in the appended revised manuscript) and removed two paragraphs (lines 513-568) in the Discussion section. However, we believe that a description of Nir2 and Nir3 protein domain functions (lines 103-113) is important to summarize what is known about the two proteins and some of this information is critical to our hypothesis that Nir3 has a greater role in regulating weak TCR signaling. Therefore, we have retained that paragraph in the introduction.

Reviewer #2 (Remarks to the Author):

The revised manuscript has been greatly improved and addresses the reviewers' concerns with significant new data.

We appreciate that Reviewer #2 believes our manuscript is improved and that we have satisfactorily responded to their comments.

Reviewer #3 (Remarks to the Author):

In this revised manuscript, the authors attempted all our suggested experiments. Especially checking levels of Nir2/3 protein in different DP1-5 subsets and mature T cells (comment # 2) and results of bone marrow chimera with OT-II mice (comment #4) were essential to justify conclusions from this study. The authors successfully carried out those experiments. The current manuscript is suitable for publication in Nature Immunology.

We are grateful that Reviewer is satisfied with our revisions and recommends it for publication in Nature Immunology.

Decision Letter, first revision:

Subject: Your manuscript, NI-A33810A

Message: Our ref: NI-A33810A

10th Oct 2022

Dear Dr. Weiss,

Thank you for your patience as we've prepared the guidelines for final submission of your Nature Immunology manuscript, "The phosphatidylinositol (PI)-transfer protein Nir3 promotes PI(4,5)P₂ replenishment in response to TCR signaling during thymocyte development and peripheral T cell survival" (NI-A33810A). Please carefully follow the step-by-step instructions provided in the attached file, and add a response in each row of the table to indicate the changes that you have made. Please also check and comment on any additional marked-up edits we have proposed within the text. Ensuring that each point is addressed will help to ensure that your revised manuscript can be swiftly handed over to our production team.

When you upload your final materials, please include a point-by-point response to any remaining reviewer comments and please make sure to upload your checklist.

If you have not done so already, please alert us to any related manuscripts from your group that are under consideration or in press at other journals, or are being written up

for submission to other journals (see: <https://www.nature.com/nature-portfolio/editorial-policies/plagiarism#policy-on-duplicate-publication> for details).

In recognition of the time and expertise our reviewers provide to Nature Immunology's editorial process, we would like to formally acknowledge their contribution to the external peer review of your manuscript entitled "The phosphatidylinositol (PI)-transfer protein Nir3 promotes PI(4,5)P₂ replenishment in response to TCR signaling during thymocyte development and peripheral T cell survival". For those reviewers who give their assent, we will be publishing their names alongside the published article.

Nature Immunology offers a Transparent Peer Review option for new original research manuscripts submitted after December 1st, 2019. As part of this initiative, we encourage our authors to support increased transparency into the peer review process by agreeing to have the reviewer comments, author rebuttal letters, and editorial decision letters published as a Supplementary item. When you submit your final files please clearly state in your cover letter whether or not you would like to participate in this initiative. Please note that failure to state your preference will result in delays in accepting your manuscript for publication.

Cover suggestions

As you prepare your final files we encourage you to consider whether you have any images or illustrations that may be appropriate for use on the cover of Nature Immunology.

Nature Immunology has now transitioned to a unified Rights Collection system which will allow our Author Services team to quickly and easily collect the rights and permissions required to publish your work. Approximately 10 days after your paper is formally accepted, you will receive an email in providing you with a link to complete the grant of rights. If your paper is eligible for Open Access, our Author Services team will also be in touch regarding any additional information that may be required to arrange payment for your article.

Please note that *Nature Immunology* is a Transformative Journal (TJ). Authors may publish their research with us through the traditional subscription access route or make their paper immediately open access through payment of an article-processing charge (APC). Authors will not be required to make a final decision about access to their article until it has been accepted. [Find out more about Transformative Journals](https://www.springernature.com/gp/open-research/transformative-journals).

If you have any questions about costs, Open Access requirements, or our legal forms, please contact ASJournals@springernature.com.

Please use the following link for uploading these materials: [REDACTED]

Best regards,

Elle Morris
Senior Editorial Assistant
Nature Immunology
Phone: 212 726 9207
Fax: 212 696 9752
E-mail: immunology@us.nature.com

On behalf of

Laurie A. Dempsey, Ph.D.
Senior Editor
Nature Immunology
l.dempsey@us.nature.com
ORCID: 0000-0002-3304-796X

Reviewer #1:

Remarks to the Author:

There are obvious improvements of the manuscript by Lu et al after revisions. For

instance, the impact of Nir3 deficiency on CD28 co-stimulation was properly assessed. Even though the Nir2 and Nir3 knockout mutants were not obtained by CRISPR/Cas9 in T cell lines, the Nir2 knockdown by siRNA in OT-I cells (stimulated with OVA and variants) shows very clearly that Nir2 is essential for T cell activation. Moreover, Nir2 knockdown causes markedly reduced Nur77 expression accompanied by reduced PI(4,5)P2 levels. The Nir2 knockdown cells had a more dramatic mutant phenotype than Nir3 knockout cells, suggesting indispensable roles of Nir2 in T cell activation. The authors answered the question concerning TCR induced phosphorylation of Nir3 by citing a published paper using the Jurkat cells expressing chimeric antigen receptor against CD19. The other experimental data added during the revision greatly improved the quality of the paper. The mutant phenotype of Nir3 knockout mice is not dramatic and primarily restricted to thymocyte development as noted by the other Reviewers. Despite this rather not spectacular phenotype, I still recommend publication of this paper because it is the first description of the role of this molecule in T cell development and for the in-depth functional analyses of animal and cellular models that the manuscript contains. Here are a few more suggestions for the authors.

1. An analysis and discussion of the TCR induced phosphorylation of Nir3 (preferably with Nir2 analyzed in parallel) will constitute an important addition to provide more mechanistic insights into the unique role of Nir3. The current data support that Nir3 plays a critical role in PI(4,5)P2 replenishment following TCR engagement in thymocytes, and its function in T cell development is probably explained the elevated expression of Nir3 during thymocyte selections during which Nir2 is not abundantly expressed. However, in spleen (Figure 1b and 1c), both Nir2 and Nir3 are present and don't really display a very dramatic difference in protein abundance. In mature T cells loss of Nir3 leads to a very obvious defect in cell survival (Figure 6a). Accordingly, phenotype observed in Nir2 knockdown cells and Nir3 knockout cells is not convincingly only explained by the abundance of Nir2 and Nir3 proteins (Reviewer Figure 2c). Their activities to replenish PIP2 might be differentially regulated as well when T cells are stimulated.
2. As the authors noted, in lymphoreplete recipient mice, most of the adoptively transferred T cells had not undergone cell division, as reflected by their largely undiluted CellTrace dye (Figure 6d-e). Can this part be replaced by results from the same mice treated with OVA to visualize in vivo expansion of Nir3 knockout cells and their controls?
3. The descriptions and comparisons about Nir2 and Nir3 on the evolutionary basis is probably too lengthy. In the current manuscript, line 103 to 113, 513 to 540, 563 to 568 provides lots of information based on the phylogenetic analysis/protein domains. However, the functional analyses of Nir3 in the other species without Nir2 is not available, and the impact of Nir2 deficiency in mice for thymocyte development is not included in this study. A more concise introduction and discussion concerning such comparisons sound more appropriate.

Reviewer #2:

Remarks to the Author:

The revised manuscript has been greatly improved and addresses the reviewers' concerns with significant new data.

Reviewer #3:

Remarks to the Author:

In this revised manuscript, the authors attempted all our suggested experiments. Especially checking levels of Nir2/3 protein in different DP1-5 subsets and mature T cells (comment # 2) and results of bone marrow chimera with OT-II mice (comment #4) were essential to justify conclusions from this study. The authors successfully carried out those experiments. The current manuscript is suitable for publication in Nature Immunology.

Final Decision Letter:

Subject: Decision on Nature Immunology submission NI-A33810B

Message: In reply please quote: NI-A33810B

Dear Dr. Weiss,

I am delighted to accept your manuscript entitled "The phosphatidylinositol (PI)-transfer protein Nir3 promotes PI(4,5)P₂ replenishment in response to TCR signaling during T cell development and survival" for publication in an upcoming issue of Nature Immunology.

Over the next few weeks, your paper will be copyedited to ensure that it conforms to Nature Immunology style. Once your paper is typeset, you will receive an email with a link to choose the appropriate publishing options for your paper and our Author Services team will be in touch regarding any additional information that may be required.

Please note that *Nature Immunology* is a Transformative Journal (TJ). Authors may publish their research with us through the traditional subscription access route or make their paper immediately open access through payment of an article-processing charge (APC). Authors will not be required to make a final decision about access to their article until it has been accepted. <https://www.springernature.com/gp/open->

research/transformational-journals">Find out more about Transformational Journals.

Authors may need to take specific actions to achieve compliance with funder and institutional open access mandates. If your research is supported by a funder that requires immediate open access (e.g. according to Plan S principles) then you should select the gold OA route, and we will direct you to the compliant route where possible. For authors selecting the subscription publication route, the journal's standard licensing terms will need to be accepted, including self-archiving policies. Those licensing terms will supersede any other terms that the author or any third party may assert apply to any version of the manuscript.

Your paper will be published online soon after we receive your corrections and will appear in print in the next available issue. Content is published online weekly on Mondays and Thursdays, and the embargo is set at 16:00 London time (GMT)/11:00 am US Eastern time (EST) on the day of publication. Now is the time to inform your Public Relations or Press Office about your paper, as they might be interested in promoting its publication. This will allow them time to prepare an accurate and satisfactory press release. Include your manuscript tracking number (NI-A33810B) and the name of the journal, which they will need when they contact our office.

About one week before your paper is published online, we shall be distributing a press release to news organizations worldwide, which may very well include details of your work. We are happy for your institution or funding agency to prepare its own press release, but it must mention the embargo date and Nature Immunology. Our Press Office will contact you closer to the time of publication, but if you or your Press Office have any enquiries in the meantime, please contact press@nature.com.

Also, if you have any spectacular or outstanding figures or graphics associated with your manuscript - though not necessarily included with your submission - we'd be delighted to consider them as candidates for our cover. Simply send an electronic version (accompanied by a hard copy) to us with a possible cover caption enclosed.

If you have not already done so, we strongly recommend that you upload the step-by-step protocols used in this manuscript to the Protocol Exchange. Protocol Exchange is an open online resource that allows researchers to share their detailed experimental know-how. All uploaded protocols are made freely available, assigned DOIs for ease of citation and fully searchable through nature.com. Protocols can be linked to any publications in which they are used and will be linked to from your article. You can also establish a dedicated page to collect all your lab Protocols. By uploading your Protocols to Protocol Exchange, you are enabling researchers to more readily reproduce or adapt the methodology you use, as well as increasing the visibility of your protocols and papers. Upload your Protocols at www.nature.com/protocolexchange/. Further information can be found at www.nature.com/protocolexchange/about .

Please note that we encourage the authors to self-archive their manuscript (the accepted version before copy editing) in their institutional repository, and in their funders' archives, six months after publication. Nature Portfolio recognizes the efforts of funding bodies to increase access of the research they fund, and strongly encourages authors to participate in such efforts. For information about our editorial policy, including license agreement and author copyright, please visit www.nature.com/ni/about/ed_policies/index.html

Sincerely,

Laurie A. Dempsey, Ph.D.
Senior Editor
Nature Immunology
l.dempsey@us.nature.com
ORCID: 0000-0002-3304-796X